# Online Linear Programming for Multi-Objective Routing in LLM Serving

**Zixi Chen** [1]  **Yinyu Ye** [2]  **Zijie Zhou** [2]

## Abstract

We study the online routing problem in large language model serving, where requests arrive sequentially and must be dispatched to parallel decode workers under tight batch-size and KV-cache constraints. Unlike widely used routing heuristics that are not tied to explicit service-level objectives (SLOs) and offer limited control over latency–throughput trade-offs, we introduce a multi-objective optimization framework that formulates routing as an online linear programming with interpretable decision rewards. We apply an efficient bid-price control policy based on the online linear programming that admits requests when their SLO-weighted benefit exceeds their shadow prices. To meet millisecond decision requirements, we develop a warm-started, projected first-order updates that track the evolving dual shadow prices online with predictable runtime. We integrate our router into the Vidur simulator and demonstrate substantial improvements over standard baselines across multiple SLO regimes, including end-to-end latency, time-to-first-token, throughput, and tail performance. A big picture from our result: a science-based approach outperforms others based on heuristics.

## 1. Introduction

The demand for large language model (LLM) services has surged dramatically in recent years, with billions of requests being processed daily (Roth, 2025). As this demand continues to grow, optimizing the efficiency of LLM serving systems has become an increasingly critical challenge (Kwon et al., 2023; Pope et al., 2023; Sheng et al., 2023; Agrawal et al., 2024b). LLM serving consists of two key stages for each request: (i) the prefill stage, where the key-value (KV) pairs of the input are generated, and (ii) the decode stage,

where the model autoregressively generates output tokens. Due to the massive volume of requests and limited computational resources, it is impossible to process all requests simultaneously. In this paper, we focus on optimizing the *router*, which dispatches incoming requests to parallel computational workers (Jain et al., 2024; Wu & Silwal, 2025; Mei et al., 2025), and we cast this control problem as a general online resource allocation task with time-coupled constraints, specialized to decode-side LLM serving.

Optimizing the router requires specifying *what* performance the system should optimize, yet real LLM services operate under heterogeneous service-level objectives (SLOs). For instance, real-time voice translation and interactive agents prioritize responsiveness, often measured by time-to-first-token, whereas chatbots emphasize end-to-end latency because users care about the time to receive a complete response; meanwhile, system operators also track throughput as a proxy for capacity and cost efficiency. These objectives are intrinsically in tension: increasing throughput typically relies on larger or more aggressive batching and higher device utilization, which raises queueing delays and degrades latencies, while prioritizing low latency often forces smaller batches or reserved capacity, reducing overall throughput and utilization. Because different products require different metric combinations and trade-offs, serving stacks frequently resort to retuning or even redesigning routing heuristics per deployment, motivating a general multi-objective control framework that can flexibly adapt routing behavior by adjusting objective weights rather than changing the router's structure.

Many production serving stacks (e.g., vLLM (Kwon et al., 2023)) and simulators (e.g., Vidur (Agrawal et al., 2024a)) adopt simple routing heuristics that fall into two broad families: *greedy load-based* rules and *randomized* rules. Representative greedy policies include join-shortest-queue and power-of-$d$ choices. While effective in classical parallel-server settings, these rules were designed for environments where job sizes are either known or homogeneous. In LLM serving, however, the relevant workload is driven by heterogeneous and uncertain decode lengths: two queues with the same number of requests can correspond to vastly different remaining work. Recent work has explored decode-length prediction (Fu et al., 2024; Zheng et al., 2023b; Qiu et al., 2024; Shahout et al., 2024), workload-aware scheduling

---

[1]School of Mathematical Sciences, Peking University [2]Department of Industrial Engineering and Decision Analytics, HKUST. Correspondence to: Zijie Zhou <jerryzhou@ust.hk>.

*Proceedings of the 43rd International Conference on Machine Learning*, Seoul, South Korea. PMLR 306, 2026. Copyright 2026 by the author(s).

(Jain et al., 2024), and multi-LLM routing (Wu & Silwal, 2025; Mei et al., 2025), yet most deployed systems still rely on heuristics that provide limited control over SLO trade-offs. Randomized baselines such as round-robin and uniform routing avoid relying on imperfect workload estimates, but they ignore the instantaneous system state and cannot adapt to congestion. More fundamentally, neither family is derived from an explicit optimization objective, providing limited and non-transparent control over trade-offs among competing SLOs—precisely the flexibility required for multi-objective serving.

The challenge of dispatching requests to capacity-constrained resources under uncertainty is a classical problem in operations research, studied extensively under the framework of online linear programming (LP) (Agrawal et al., 2014; Li et al., 2020; Li & Ye, 2022) and bid-price control in revenue management (Talluri & Van Ryzin, 1998; Akan & Ata, 2009). A key insight from this literature is that good online decisions should balance the *immediate benefit* of serving a request against the *opportunity cost* of consuming scarce capacity that may be needed for future arrivals. LP (Luenberger et al., 1984) duality provides a principled way to quantify these opportunity costs through *shadow prices*. In LLM serving, the scarce resources are batch slots and KV-cache memory on each device across future time steps—resources that are *time-coupled* because a request admitted now occupies capacity throughout its decode duration. Beyond constraints, the *objectives* themselves require careful design: aggregate SLOs such as tail latency are statistical properties that do not directly decompose into per-request rewards. A key modeling contribution of our work is a reward structure that converts these aggregate metrics into per-request incentives while ensuring that scheduling is usually beneficial. This combination of time-coupled constraints and SLO-decomposable objectives motivates our online LP framework.

To our knowledge, no prior work provides a multi-objective optimization framework for LLM routing that (i) explicitly models time-coupled KV-cache and batch-size constraints inherent to autoregressive decoding, (ii) decomposes aggregate SLOs—including tail latency—into per-request rewards suitable for online optimization, and (iii) derives computationally efficient policies grounded in LP duality. Our work bridges this gap by bringing operations research methodology to LLM serving. Specifically, we make the following contributions:

1. **Multi-objective optimization framework.** We formulate LLM routing as an online linear program with time-coupled batch and KV-cache constraints. The objective function explicitly incorporates multiple SLOs—throughput, end-to-end latency, time-to-first-token, and tail latency—through interpretable weights that opera-

tors can adjust to match product requirements.

2. **Efficient bid-price control algorithm.** We derive a bid-price policy where each routing decision compares the request's SLO-weighted benefit against its shadow-price cost. To meet millisecond decision latency requirements, we develop warm-started projected gradient updates that track evolving dual prices online without solving an LP at each step.

3. **Interpretable trade-off control.** Through the LP formulation, operators gain transparent control over SLO trade-offs: adjusting objective weights systematically shifts performance along the Pareto frontier, and shadow prices reveal which resources (batch slots vs. KV-cache) are bottlenecks under different workloads.

We validate our framework through extensive simulation experiments using the Vidur simulator (Agrawal et al., 2024a), demonstrating substantial improvements over standard routing heuristics across multiple SLO regimes. While full integration into production serving systems remains future work, our results establish the potential of optimization-based routing and provide a principled foundation for the systems community. We also discuss the structural insights that emerge from viewing LLM routing through an optimization lens—insights that may guide future system design even beyond our specific algorithm.

## 2. Mathematical Model

In this section, we present a mathematical model for multi-worker LLM serving under *Prefill/Decode (PD) disaggregation* (Zhong et al., 2024) and *continuous batching*. A discussion of how the model and algorithms are adapted under PD mixing is deferred to Appendix A.

**Time, devices, and requests.** We consider a finite horizon of discrete periods $k = 1, \ldots, T$ and a set of decode devices indexed by $g \in [G]$. One period corresponds to one decode iteration of a batched decode kernel on a device. An instance of requests $\mathcal{J}$ arrives sequentially over the horizon. Each request $j \in \mathcal{J}$ completes its prefill stage and arrives at time $t_j \in \{1, \ldots, T\}$ with: (i) a known prefill footprint $s_j$ (e.g., the number of KV-cache token slots already materialized by prefill), and (ii) an unknown decode length $o_j$ (the number of decode tokens required to complete the response). In practice, $o_j$ can be predicted using established techniques (e.g., (Zheng et al., 2023b; Qiu et al., 2024; Fu et al., 2024; Chen et al., 2024; Shahout et al., 2024)); we denote the predicted decode length by $\hat{o}_j$, which may be inaccurate.

**Router-side buffering.** After completing prefill, each request $j$ enters a *central router queue* at time $t_j$. At the beginning of each period $k$, the router observes the current state

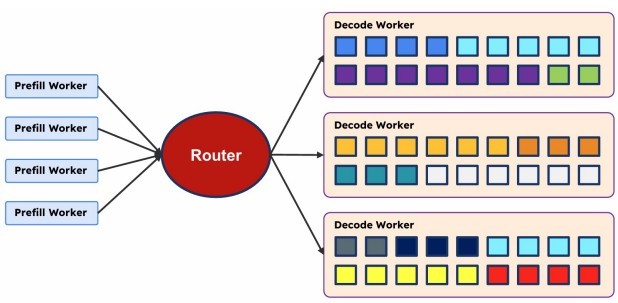

*Figure 1.* Illustration of Prefill/Decode (P/D) disaggregation with continuous batching. Each small box denotes a token, and colors indicate distinct requests. Within each decode worker, active requests are processed token-by-token in a continuously updated batch as completed requests are replaced by newly admitted ones.

of each decode device, and may *release* a subset of queued requests to devices. This design is supported by recent versions of the vLLM engine, which implement router-side buffering (vLLM Project, 2025; vLLM Roadmap, 2025).

In this paper, we focus on a *release-when-startable* design: the router dispatches a request to a device only if the request can be admitted into that device's active decode batch in the same period. In implementation, one may still maintain short per-device FIFO queues, but under our policy these queues remain small.

**Routing and admission decisions.** For each request $j \in \mathcal{J}$, device $g \in [G]$, and period $\tau \in \{t_j, \ldots, T\}$, we define a **routing/admission decision variable**

$$x_{j,g,\tau} \in [0,1],$$

interpreted as: "request $j$ is released from the router queue and admitted to device $g$ at the beginning of period $\tau$." In an integral schedule, $x_{j,g,\tau} \in \{0,1\}$ and at most one pair $(g,\tau)$ is selected for each request: $\sum_{g \in [G]} \sum_{\tau=t_j}^{T} x_{j,g,\tau} \leq 1, \forall j \in \mathcal{J}$.

**Continuous batching within a decode device.** Within each decode device, we model the classical continuous batching mechanism used by vLLM (Kwon et al., 2023). At each period, device $g$ executes one batched decode iteration that processes *one token* for every *active* request currently in its batch. A request $j$ requires $o_j$ such iterations to complete. As soon as an active request finishes, the device can admit new requests into the batch in subsequent iterations, thereby maintaining a continuously updated batch (see Figure 1).

**KV-cache memory accounting.** During decoding, the KV-cache footprint of request $j$ increases by (approximately) one token slot per decoded token. We represent the KV-cache usage of request $j$ after $\theta$ decoded tokens by

the kernel $m_j^{\mathrm{mem}}(\theta)$:

$$m_j^{\mathrm{mem}}(\theta) = \begin{cases} s_j + \theta + 1, & 0 \leq \theta < o_j, \\ 0, & \text{otherwise,} \end{cases} \quad (1)$$

where $\theta$ is the number of decode tokens already generated for request $j$. (Equivalently, while $j$ is active, its memory footprint equals its prefill footprint plus its current decoded length.)

**Device feasibility constraints.** A device cannot batch arbitrarily many requests due to two common constraints:

**(i)** *Batch-size constraint:* at any period, each decode device can host at most $B$ active requests in the batch.

**(ii)** *KV-cache capacity:* at any period, the total KV-cache usage of all active requests on a decode device cannot exceed a budget $M$; otherwise, the system must swap requests out and stall compute. In our model, we enforce routing/scheduling decisions that respect this capacity.

Here, treating one decode iteration as one time period is reasonable since under PD disaggregation, decode workers process only decode iterations, eliminating the primary source of iteration-time variability identified in non-disaggregated systems. In a decode-only regime, each iteration is memory-bandwidth-bound: the dominant cost is loading model weights, which is constant regardless of batch composition. The attention component grows linearly with the total cached KV tokens, but profiling reported by Jain et al. (2024) shows this slope is roughly $10\times$ smaller than the corresponding prefill slope, indicating that decode iteration time varies slowly with batch state.

**Objective Metrics.** LLM serving is naturally multi-objective. We consider the following standard performance metrics.

**Service-level metrics.**

**(1)** *Average/Tail end-to-end latency (EEL):* Let $c_j$ denote the period when request $j$ completes (i.e., its final token is processed). The end-to-end latency of request $j$ is $c_j - t_j$. The average EEL is

$$\frac{1}{|\mathcal{J}|} \sum_{j \in \mathcal{J}} (c_j - t_j).$$

For a tail level $x\%$, the $x\%$ tail EEL is the $x\%$ quantile of the multiset $\{c_j - t_j : j \in \mathcal{J}\}$.

**(2)** *Average/Tail time-to-first-token (TTFT):* Let $\tau_j$ denote the period when request $j$ starts decoding, i.e., when

its *first* output token is generated. The TTFT of request $j$ is $\tau_j - t_j$. The average TTFT is

$$\frac{1}{|\mathcal{J}|} \sum_{j \in \mathcal{J}} (\tau_j - t_j),$$

and the $x\%$ tail TTFT is the $x\%$ quantile of $\{\, \tau_j - t_j : j \in \mathcal{J} \,\}$.

*Remark* 2.1. We do not explicitly report *time-per-output-token* (TPOT) since TPOT can be recovered as end-to-end latency minus TTFT and normalized by $o_j$.

**System-level metrics.**

(3) *Throughput:* Let $N^{\text{tok}}(T)$ be the total number of decode tokens processed over periods $1, \ldots, T$ across all devices. The (average) token throughput over the horizon is

$$N^{\text{tok}}(T)/T.$$

(4) *Queries-per-second (QPS):* Let $N^{\text{req}}(T)$ be the number of requests that complete by time $T$. The (average) QPS over the horizon is

$$N^{\text{req}}(T)/T.$$

# 3. Methodology and Algorithm Design

In this section, we present an online linear programming-based algorithm designed to optimize multiple objectives in the LLM serving process.

## 3.1. Online Linear Programming Formulation

At each decoding step $k \in \{1, 2, \ldots, T\}$, we define the set of requests that have not yet started decoding (i.e., they are waiting in the queue) as $\mathcal{J}_k^{(wait)}$. The activity indicator for decoding steps is defined as $a_j(\theta) = \mathbf{1}\{0 \leq \theta < \hat{o}_j\}$, where $\theta$ represents the number of decoding steps processed by request $j$ since its start time. If a request starts at time $\tau$, then $a_j(k - \tau)$ indicates whether the request is currently being processed at time $k$ (taking values 1 or 0 depending on whether it is active and within the batch size limit). The memory usage of the request follows the kernel $m_j^{\text{mem}}(\theta)$ defined in (1).

Given that each worker is subject to both a batch size constraint and a memory usage constraint, we define a constraint-type index $u \in \{\mathsf{bat}, \mathsf{mem}\}$, where:

- $\mathsf{bat}$ represents the per-period batch size constraint;

- $\mathsf{mem}$ represents the per-period memory usage constraint.

We define the set of device-time-constraint coordinates as

$$\mathcal{I} := [G] \times [T] \times \{\mathsf{bat}, \mathsf{mem}\}, \quad i = (g, k, u) \in \mathcal{I}.$$

For each device-time-constraint coordinate $i = (g, k, u) \in \mathcal{I}$ and request $j$, the scheduling action column coefficient is defined as

$$a_{i,(j,g',\tau)} = \begin{cases} a_j(k - \tau), & \text{if } u = \mathsf{bat}, \ g' = g, \\ m_j^{\text{mem}}(k - \tau), & \text{if } u = \mathsf{mem}, \ g' = g, \\ 0, & \text{otherwise.} \end{cases}$$

For each device-time-constraint coordinate $i = (g, k, u)$, we set

$$b_{(g,k,\mathsf{bat})} = B_{g,k}, \quad b_{(g,k,\mathsf{mem})} = M_{g,k},$$

where $B_{g,k}$ represents the remaining batch size capacity (i.e., the difference between the total batch size limit $B$ and the number of active requests) of device $g$ at time $k$, and $M_{g,k}$ represents the remaining KV memory usage capacity of device $g$ at time $k$. At decision step $k \in [T]$, only requests that have not yet started (i.e., those in $\mathcal{J}_k^{(wait)}$) can be scheduled. We define the action (column) set as

$$\mathcal{A}_k := \{(j, g, k) : j \in \mathcal{J}_k^{(wait)}, g \in [G]\}.$$

The system-level constraints for each device $g$ at time $k$ can be written as

$$\sum_{(j,g,k) \in \mathcal{A}_k} a_{i,(j,g,k)} x_{j,g,k} \leq b_i, \quad \forall i = (g, k, u) \in \mathcal{I}.$$

At step $k$, we make decisions only for the waiting requests $j \in \mathcal{J}_k^{(wait)}$ about which device and time to start processing. Although $x_{j,g,k}$ represents a one-time start decision, it induces resource usage across multiple future steps $t \geq k$ through the age-dependent kernels: $a_j(t - k)$ for batch occupancy and $m_j^{\text{mem}}(t - k)$ for KV memory usage. A request automatically stops contributing to the resource constraints once it finishes decoding, as $a_j(\theta) = 0$ and $m_j^{\text{mem}}(\theta) = 0$ when $\theta \geq \hat{o}_j$. The right-hand side $b_i$ represents the remaining capacity after subtracting the load of previously started requests, ensuring that newly admitted requests do not violate batch size or memory constraints on any device at any future decoding step.

**Linear Multiple Objective and Decision Reward** To formulate this problem as a linear program, we assign a reward to each decision $(j, g, k)$, corresponding to assigning request $j$ to device $g$ at time $k$. The reward function is given

by

$$r_{j,g,k} = \underbrace{\alpha \cdot \mathtt{Thr}_{j,g,k}}_{\text{throughput}} - \underbrace{\beta \cdot \mathtt{EEL}_{j,g,k}}_{\text{average end-to-end latency}}$$
$$- \underbrace{\gamma \cdot \mathtt{TTFT}_{j,g,k}}_{\text{average TTFT}} + \underbrace{\sigma \cdot \mathtt{QPS}_{j,g,k}}_{\text{queries per second}}$$
$$- \underbrace{\zeta_1 \cdot \mathtt{Tail\ EEL}_{j,g,k}}_{\text{tail end-to-end latency}} - \underbrace{\zeta_2 \cdot \mathtt{Tail\ TTFT}_{j,g,k}}_{\text{tail TTFT}},$$

where $\alpha, \beta, \gamma, \sigma, \zeta_1, \zeta_2 \geq 0$ are parameters representing the relative importance of each objective. These parameters should be selected according to the specific product and its purpose, as different products may prioritize different metrics.

Next, we decompose each term in the reward to better understand how the metrics contribute to the overall decision.

For system-level metrics, the throughput contribution of an action is given by the number of tokens processed during the time horizon $[0, T]$. If a request is assigned, its true contribution to throughput is the minimum of its predicted decode length and the remaining time steps: $\mathtt{Thr}_{j,g,k} = \min\{\hat{o}_j, T-k+1\}$. For QPS, the contribution of an action is simply 1, as each assigned request contributes one processed query: $\mathtt{QPS}_{j,g,k} = 1$.

For service-level metrics, consider the contribution of each action to the average latency. Over the time horizon $[0, T]$, we have:

$$\mathtt{EEL}_{j,g,k} = \min\{k+\hat{o}_j, T+1\} - t_j, \quad \mathtt{TTFT}_{j,g,k} = k - t_j.$$

However, although we seek to minimize latency, we **cannot allow it to become negative**. If we provide negative penalties for latency, the router may choose not to assign requests in order to avoid incurring a penalty, especially as the time horizon $T$ approaches its end. To avoid this issue, we observe that both end-to-end latency and TTFT can be written as:

$$\sum_{s=t_j}^{T} \mathbf{1}\{\text{not finished by } s\} = (T-t_j+1) - \big(T-(k+\hat{o}_j-1)\big)_+, \tag{2}$$

and

$$\sum_{s=t_j}^{T} \mathbf{1}\{\text{not started by } s\} = (T-t_j+1) - (T-k+1), \tag{3}$$

respectively. In both (2) and (3), the term $(T - t_j + 1)$ is independent of the decision process, and we can eliminate this term. The only penalty related to the decision is $-\big(T - (k+\hat{o}_j-1)\big)_+$ and $-(T-k+1)$, which makes the penalties negative and the reward positive.

We also introduce an interesting and simple method to decompose statistical tail latency into an instant reward for the action. Taking tail TTFT as an example, suppose the service-level constraint is that the tail TTFT must be bounded by $t'$. We define an indicator reward as

$$\mathtt{Tail\ TTFT}_{j,g,k} = \mathbf{1}\{k - t_j < t'\}.$$

This guarantees that if the TTFT of a request is within the desired bound $t'$, a reward of 1 unit is added. To enforce a stricter tail TTFT requirement (e.g., the 99th percentile), we set a larger weight for $\zeta_2$, which increases the weight on tail TTFT in the reward and drives decisions to prioritize it. Thus, the weights associated with the tail latency metrics, $\zeta_1$ and $\zeta_2$, not only determine the relative importance of tail latency in comparison to other objectives, but also directly map the target quantiles for the tail latency. A detailed guideline on how to select these parameters will be provided in Section 4.

Finally, the overall positive reward for action $(j, g, k)$ is written as:

$$r_{j,g,k} = \underbrace{\alpha \cdot \min\{\hat{o}_j, T-k+1\}}_{\text{throughput}} + \underbrace{\beta \cdot \big(T-(k+\hat{o}_j-1)\big)_+}_{\text{end-to-end latency "saved steps"}}$$
$$+ \underbrace{\gamma \cdot (T-k+1)}_{\text{first-token "saved steps"}} + \underbrace{\sigma}_{\text{queries per second}}$$
$$+ \underbrace{\zeta_1 \cdot \mathbf{1}\{k + \hat{o}_j - t_j < t'_1\}}_{\text{indicator reward for tail EEL}} + \underbrace{\zeta_2 \cdot \mathbf{1}\{k - t_j < t'_2\}}_{\text{indicator reward for tail TTFT}}.$$

The corresponding online linear programming formulation at time $k$ is:

$$\max_x \sum_{(j,g,k)\in\mathcal{A}_k} r_{j,g,k} x_{j,g,k}$$
$$\text{s.t.} \sum_{(j,g,k)\in\mathcal{A}_k} a_{i,(j,g,k)} x_{j,g,k} \leq b_i, \quad \forall i \in \mathcal{I}, \tag{4}$$
$$\sum_{g\in[G]} x_{j,g,k} \leq 1, \quad \forall j \in \mathcal{J}_k^{(wait)}. \tag{5}$$

### 3.2. Duality, Bid-Price Control, and Stochastic Program

The per-step LP provides a clean abstraction of the routing decision at time $k$. However, in an LLM serving system the decision loop must operate at millisecond scale, so repeatedly solving an LP to optimality at every step is impractical. We therefore leverage LP duality to derive a *bid-price control* rule (Talluri & Van Ryzin, 1998; Akan & Ata, 2009; Agrawal et al., 2014; Li & Ye, 2022): instead of re-optimizing from scratch, we update the dual prices (resource shadow costs) and make decisions by comparing each request's reward against its implied resource cost.

**Dual formulation and bid prices.** Introduce dual variables $p_i \geq 0$ for the packing constraints (4) and $y_j \geq 0$ for

the per-request constraint (5). For the per-step LP at time $k$, the standard dual is

$$\min_{p\geq 0,\, y\geq 0} \quad \sum_{i\in\mathcal{I}} b_i\, p_i + \sum_{j\in\mathcal{J}_k^{(wait)}} y_j$$

$$\text{s.t.} \quad \sum_{i\in\mathcal{I}} a_{i,(j,g,k)}\, p_i + y_j \geq r_{j,g,k}, \; \forall (j,g,k)\in\mathcal{A}_k.$$

The dual variables admit concrete operational meaning. For each resource-time coordinate $i = (g,k,u)\in\mathcal{I}$, the dual variable $p_i$ represents the *shadow price* of resource type $u$ on device $g$ at time $k$: it quantifies the marginal value of one additional unit of batch capacity (when $u = \mathsf{bat}$) or KV-cache memory (when $u = \mathsf{mem}$). High shadow prices indicate anticipated congestion on that resource, while low prices indicate spare capacity. For each request $j$, the dual variable $y_j$ represents the *option value* of request $j$—the maximum net benefit (SLO reward minus imputed resource cost) achievable by routing $j$ to its best device.

For a fixed price vector $p$, the optimal slack variable is

$$y_j^*(p) = \left[ \max_{g\in[G]} \left\{ r_{j,g,k} - \sum_{i\in\mathcal{I}} a_{i,(j,g,k)}\, p_i \right\} \right]_+.$$

Substituting $y^*(p)$ yields the reduced dual objective

$$\min_{p\geq 0} \quad \sum_{i\in\mathcal{I}} b_i p_i + \sum_{j\in\mathcal{J}_k^{(wait)}} \left[ \max_{g\in[G]} \left\{ r_{j,g,k} - a_{(j,g,k)}^\top p \right\} \right]_+,$$

where $a_{(j,g,k)}^\top p := \sum_{i\in\mathcal{I}} a_{i,(j,g,k)} p_i$ and $a_{(j,g,k)} := (a_{i,(j,g,k)})_{i\in\mathcal{I}} \in \mathbb{R}_+^{|\mathcal{I}|}$. The quantity $a_{(j,g,k)}^\top p$ is the *imputed resource cost* of admitting $(j,g,k)$ under shadow prices $p$. Accordingly, the *bid-price margin*

$$\Delta_j(g) := r_{j,g,k} - a_{(j,g,k)}^\top p$$

measures the net benefit of assigning request $j$ to device $g$ at time $k$ after accounting for resource opportunity costs.

**Sample-Average Approximation (SAA)-style price learning.** The reduced dual objective depends on the random stream of arrivals and on predicted decode length profiles. We therefore view $p$ as the solution to an underlying stochastic program and estimate it from historical observations. Let Hist denote a history set of previously observed pairs $(r^\star, a^\star)$, where each $(r^\star, a^\star)$ corresponds to a candidate action generated in the past. At decision step $k$, we compute prices by minimizing the SAA:

$$f_k(p) = d_k^\top p + \frac{1}{\max\{1, |\mathsf{Hist}|\}} \sum_{(r^\star, a^\star)\in\mathsf{Hist}} \left( r^\star - (a^\star)^\top p \right)_+,$$

(6)

where $d_k := \frac{b^{(k-1)}}{\max\{1, \widehat{n}_{\mathrm{remain}}(k)\}}$. Here, to simplify the notation, we denote $b^{(k)} := (b_i^{(k)})_{i\in\mathcal{I}} \in \mathbb{R}_+^{|\mathcal{I}|}$, and $b^{(k-1)}$ is the residual-capacity vector before making decisions at step $k$, and $\widehat{n}_{\mathrm{remain}}(k)$ is an forecast of the expected number of remaining admission opportunities.

**Bid-price control online decision.** Given the price vector $p_k$ learned from history, a natural bid-price rule is to accept a request $j$ if its best margin, $\Delta_j(g) := r_{j,g,k} - a_{(j,g,k)}^\top p_k$, is positive. However, at a single step $k$ there may be many waiting requests, and admitting *all* actions with positive margins can violate residual constraints. Moreover, rewards and resource profiles are based on predictions (e.g., $\hat{o}_j$), which introduces estimation error. To robustly enforce feasibility, we (i) compute bid price margins for all candidates, (ii) rank candidates by margin, and (iii) greedily admit the highest-margin actions subject to the residual constraints and the one-assignment-per-request rule. The pseudocode can be found in Algorithm 1 in Appendix B.1.

### 3.3. Projected Online Dual Gradient Descent

Algorithm 1 computes bid prices by solving the convex dual subproblem:

$$p_k \in \arg\min_{p\geq 0} f_k(p),$$

(7)

where $f_k(p)$ is defined in Equation (6). While (7) is substantially smaller than the primal LP, solving it to optimality at every decision step can still be nontrivial at millisecond-scale latency. We therefore suggest a method based on online gradient descent (Zinkevich, 2003; Sun et al., 2020) that updates $p_k$ from $p_{k-1}$. This is motivated by the fact that the dual objective evolves smoothly over time: between steps $k-1$ and $k$, the history set and resource capacities change incrementally.

**Warm-started projected mini-batch updates.** At step $k$, we initialize $p^{(0)} \leftarrow p_{k-1}$ and perform a small fixed number $K$ of projected subgradient steps using a mini-batch $\mathcal{B}_k \subseteq \mathsf{Hist}_k$ sampled uniformly without replacement:

$$p^{(s+1)} \leftarrow \Pi_{\mathbb{R}_+^{|\mathcal{I}|}} \left( p^{(s)} - \eta_k\, \widehat{g}_k(p^{(s)}) \right),$$

(8)

where

$$\widehat{g}_k(p) := d_k - \frac{1}{|\mathcal{B}_k|} \sum_{(r,a)\in\mathcal{B}_k} \mathbf{1}\{r - a^\top p > 0\}\, a,$$

for $s = 0, 1, \ldots, K-1$, and output $p_k := p^{(K)}$. Here $\Pi_{\mathbb{R}_+^{|\mathcal{I}|}}$ denotes projection onto the nonnegative orthant, implemented entrywise as $(\cdot)_+$. This first-order update has short and predictable runtime (controlled by $K$ and $|\mathcal{B}_k|$) and avoids repeatedly solving (7).

**Implementation details.** In our LLM serving instantiation, the constraint index $i = (g, k, u)$ induces a structured price vector with two tensors (batch-occupancy and memory) over $(g, k)$, and we compute $a^\top p$ efficiently using only the active time slots of each candidate column. We also apply mild normalization to keep the hinge-loss gradients numerically stable. The details are provided in Appendix B.

**Computational overhead and scaling.** The online routing path does not require solving a full primal LP at any step. At each period, the dual variable has dimension $2GT$, but updates are highly structured. Each routing step performs only $K$ warm-started projected gradient updates over a mini-batch sampled from the action history, and the dot product $\mathbf{a}^\top \mathbf{p}$ for any candidate column touches only the *active* future slots of that request, so the per-step cost scales with the number of active candidates times their average decode length rather than the full $GT$ horizon. The state itself is summarized by $O(GT)$ memory, and the per-step time complexity is $O(KGH)$ where $H$ is the effective look-ahead horizon. Both quantities are linear in cluster size, so the method scales gracefully to larger deployments—each worker's price tensor can also be updated in parallel since they are decoupled in the gradient step. In our current implementation, on an Intel i9-13980HX CPU, the routing computation takes approximately 1–2ms per period, comfortably within the 10–30ms wall-clock time of a single decode iteration on an A100 GPU. This confirms that the method meets the millisecond-scale decision requirements of online LLM serving and is unlikely to be the bottleneck even at 64+ workers.

# 4. Numerical Experiments

In this section, we present the results of our numerical experiments. Due to lack of GPU resources, we conducted our experiments using the Vidur simulator (Agrawal et al., 2024a), which models realistic LLM serving behavior including continuous batching and KV-cache management. While simulation cannot capture all aspects of real deployments, it allows controlled study of routing policies across diverse scenarios. Our routing algorithm has been integrated into the Vidur simulator, and the code is available at `https://github.com/qqwetidx/Online-Linear-Programming-for-Vidur`.

## 4.1. Experimental Setup

We simulate a scenario with four parallel NVIDIA A100 GPUs. To evaluate the performance of our proposed routing algorithm, we consider two types of request distributions:

- **Real Data:** We test on a conversational dataset provided by (Zheng et al., 2023a), which is pub-

licly available at `https://huggingface.co/datasets/lmsys/lmsys-chat-1m`. The real-world results are summarized in the main text, with additional experiments available in Appendix C.1.

- **Synthetic Data:** We also simulate requests with varying prefill-to-decode (P/D) ratios, specifically: (i) P/D ratio 1:4 (more prefill than decode); (ii) P/D ratio 4:1 (more decode than prefill). These settings simulate different workload patterns, ranging from long prefill phases to long decoding phases. The results for this synthetic dataset are presented in Appendix C.2.

For the real dataset, the request arrival process is modeled as a stationary Poisson process, with an arrival rate $\lambda \in \{0.4, 0.5\}$. We observe that a rate of $\lambda = 0.4$ ensures system stability, while $\lambda = 0.5$ leads to a slight overload. Each arriving request is randomly sampled from the dataset.

## 4.2. Baseline Algorithms and Parameter Settings

We compare the performance of our algorithm against all routing heuristics which are default in the Vidur simulator: (i) **Round Robin:** Requests are cyclically assigned to the GPUs in a fixed order, regardless of the current system state. (ii) **Least Outstanding Request (LOR):** This policy assigns requests to the GPU with the shortest queue. (iii) **Random:** Requests are randomly assigned to any of the four GPUs with equal probability $1/4$. (iv) **Power-of-2 Choices:** For each incoming request, two GPUs are sampled uniformly at random and the request is assigned to the one with fewer outstanding requests. This is a widely used randomized load-balancing heuristic known to improve over both Random and pure shortest-queue rules in classical parallel-server settings.

For our routing algorithm, we conduct experiments with various configurations to examine the effects of prediction accuracy and different objective weightings:

- **Prediction Accuracy:** We explore two scenarios for the predicted decode length $\hat{o}_j$: (i) Perfect Prediction: In some experiments, we assume perfect prediction, where $\hat{o}_j = o_j$. (ii) Imperfect Prediction: In other experiments, we model a more realistic scenario with 20% prediction error, where $\hat{o}_j \sim \text{Unif}(0.8 \cdot o_j, 1.2 \cdot o_j)$.

- **Objective Weights:** We vary the weights $\alpha, \beta, \gamma, \sigma, \zeta_1, \zeta_2$ for the different service-level objectives (SLOs) to observe the algorithm's performance under different priorities.

## 4.3. Results

We report sample results on the real-data setting described in Section 4. Additional experiments across parameter com-

*Table 1.* Relative improvement (%) over Round-Robin with noisy decode-length prediction $\hat{o}_j \sim \text{Unif}(0.8o_j, 1.2o_j)$.

| Method | Avg EEL↓ | P95 EEL↓ | P99 EEL↓ | Thr.↑ | SLO↓ |
|---|---|---|---|---|---|
| LOR | 0.67 | 4.01 | 6.19 | 0.53 | 0.98 |
| Power-of-2 | 1.30 | 3.84 | 6.06 | 0.73 | 1.84 |
| **Ours** | **45.75** | **42.49** | **25.85** | **0.90** | **11.10** |

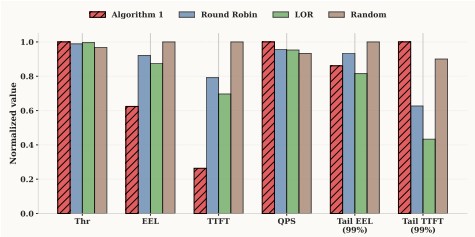

*(a)* $\lambda = 0.5$; $(\alpha, \beta, \gamma, \sigma, \zeta_1, \zeta_2) = (0, 1, 1, 0, 0, 0)$; $\hat{o}_j = o_j$

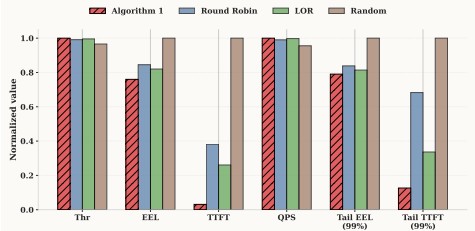

*(b)* $\lambda = 0.4$; $(\alpha, \beta, \gamma, \sigma, \zeta_1, \zeta_2) = (0, 0, 0, 0, 1, 1)$; $\hat{o}_j \sim \text{Unif}(0.8o_j, 1.2o_j)$

*Figure 3.* Real-data comparison illustrating how objective weights shift performance trade-offs.

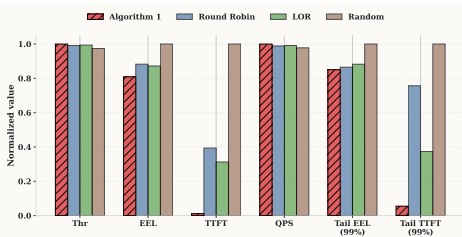

*(a)* $\lambda = 0.4$; $(\alpha, \beta, \gamma, \sigma, \zeta_1, \zeta_2) = \left(\frac{1}{135}, \frac{1}{400}, \frac{1}{400}, 1, 1, 1\right)$; $\hat{o}_j = o_j$

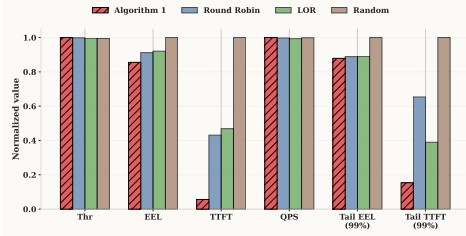

*(b)* $\lambda = 0.4$; $(\alpha, \beta, \gamma, \sigma, \zeta_1, \zeta_2) = \left(\frac{1}{27}, \frac{1}{2048}, \frac{1}{2048}, 5, 1, 1\right)$; $\hat{o}_j \sim \text{Unif}(0.8o_j, 1.2o_j)$

*Figure 2.* Real-data comparison between routing policies under two representative multi-objective settings.

binations are deferred to Appendix C.1.

**Real-data performance under representative multi-objective settings.** Figure 2 presents two representative configurations: one with perfect decode-length prediction ($\hat{o}_j = o_j$) and one with noisy prediction. In both cases, we choose a relatively large QPS weight $\sigma$ to discourage idling and sustain throughput under load, and we set $(\zeta_1, \zeta_2)$ large to emphasize tail-latency control. Across both prediction regimes, our router achieves substantially smaller TTFT and tail TTFT than all baselines. Importantly, from Table 1, the advantage persists under noisy $\hat{o}_j$, indicating that the policy is robust to moderate prediction error.

**Controlling trade-offs by adjusting objective weights.** A central advantage of our LP-based formulation is that it exposes an explicit, interpretable knob for trading off competing SLOs. Figure 3 illustrates this effect by considering two extreme weight choices. In the first setting, we emphasize average latency by setting $(\alpha, \beta, \gamma, \sigma, \zeta_1, \zeta_2) = (0, 1, 1, 0, 0, 0)$, which substantially reduces average end-to-end latency and average TTFT, but can worsen tail metrics

because the policy is not incentivized to protect the slowest requests. In the second setting, we emphasize tail control via $(\alpha, \beta, \gamma, \sigma, \zeta_1, \zeta_2) = (0, 0, 0, 0, 1, 1)$, which markedly improves tail end-to-end latency and tail TTFT.

**Robustness to non-stationary workloads.** A natural concern with any router that learns from historical arrivals is whether it can adapt when the workload distribution shifts. To stress-test our policy, we run a rate-shift experiment in which the Poisson arrival rate switches from $\lambda = 0.4$ to $\lambda = 0.6$ at the simulation midpoint, simulating a sudden load spike. The router is *not* informed of either the shift point or the new rate. To handle this online, we equip the SAA-style price update with a simple rolling-window mechanism: we monitor the empirical arrival rate over a sliding window and, when a statistically significant increase is detected, we discard stale history samples and re-estimate using only recent observations. This integrates naturally with the warm-started projected gradient descent, since the dual prices are already updated incrementally rather than recomputed from scratch. Table 2 reports the relative improvement of each method over Round Robin under this non-stationary protocol with noisy decode-length prediction.

## 5. Discussion

Beyond algorithmic improvements, our LP framework provides structural insights about LLM routing and suggests directions for future system design.

*Table 2.* Relative improvement (%) over Round Robin under a non-stationary rate-shift workload (Poisson rate switches from $\lambda = 0.4$ to $\lambda = 0.6$ at the midpoint; router uninformed of the shift). Decode lengths are noisy: $\hat{o}_j \sim \text{Unif}(0.8o_j, 1.2o_j)$.

| Method | Avg EEL↓ | P95 EEL↓ | P99 EEL↓ | Thr. ↑ | SLO Viol.↓ |
|---|---|---|---|---|---|
| LOR | 1.47 | 6.72 | 9.67 | 1.48 | 3.92 |
| Power-of-2 | 1.80 | 6.35 | 8.28 | 1.39 | 3.28 |
| **Ours** | **44.81** | **45.69** | **31.15** | **1.93** | **14.57** |

**What the framework reveals.** First, shadow prices serve as *congestion signals*: they reveal which resources—batch slots versus KV-cache memory—are bottlenecks under different workloads. In decode-heavy scenarios with long output sequences, memory prices dominate; in high-arrival-rate scenarios, batch prices dominate. This diagnostic information is unavailable from heuristic routers. Second, the framework exposes the *trade-off structure* between competing SLOs: varying objective weights traces the Pareto frontier between throughput and latency, enabling systematic capacity planning rather than ad-hoc tuning. Third, our experiments demonstrate *robustness to prediction error*: the bid-price mechanism degrades gracefully under inaccurate decode-length predictions because prices adapt online, whereas fixed heuristics cannot adjust to prediction errors.

**Implications for system design.** Our framework suggests several directions for practitioners. The shadow-price mechanism could be integrated with real-time telemetry (queue depths, memory utilization) for adaptive routing that responds to transient congestion. During overload, the framework naturally supports SLO-aware admission control: requests whose SLO contribution falls below their resource cost can be delayed or rejected, prioritizing high-value traffic. The formulation also extends naturally to heterogeneous GPU fleets by specifying device-specific capacity constraints $(B_g, M_g)$, enabling unified routing across mixed hardware.

**When does optimization help most?** Our experiments suggest the framework provides the largest benefits when (i) decode lengths are long and variable, so time-coupled constraints bind; (ii) multiple SLOs are genuinely in tension; and (iii) the system operates near full capacity. For light loads or very short decodes, simple heuristics often suffice—a regime our framework identifies through near-zero shadow prices. This characterization helps practitioners decide when optimization-based routing is worth the added complexity.

## 6. Limitations and Future Directions

Our validation relies on simulation using the Vidur framework; real system implementation would face additional challenges including integration with serving engines (e.g., vLLM), handling of preemption and KV-cache swapping, and coordination overhead in distributed deployments. Addressing these engineering challenges and validating performance on production workloads is an important direction for future work.

The current framework also assumes homogeneous request priorities and does not explicitly model fairness across users or request classes. Extending the objective to incorporate priority tiers, fairness constraints, and load-balancing considerations—while preserving computational efficiency—is a natural next step. More broadly, we hope this work demonstrates the value of bringing principled optimization methodology to LLM serving and provides a foundation for future research at the intersection of operations research and machine learning systems.

## Impact Statement

From a methodological perspective, our work establishes the first multi-objective optimization framework for LLM request routing, formulating the problem as online linear programming with time-coupled capacity constraints. This provides a principled bridge between the operations research community—where online resource allocation and bid-price control are well-studied—and the LLM serving systems community, where routing decisions have largely relied on heuristics designed for classical settings. Beyond any specific algorithmic improvements, the framework offers interpretable tools for systematic policy design: objective weights that transparently control SLO trade-offs, and shadow prices that diagnose capacity bottlenecks. We hope this work demonstrates the value of bringing optimization methodology to emerging LLM systems challenges and provides a foundation for future research at this intersection.

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

# A. Supplementary Materials for Section 2

**Overview.** Section 2 considers PD disaggregation: prefill is completed upstream and each decode device only performs decoding. We now sketch an extension to *PD mixing*, where each device may process both prefill and decode and continuous batching interleaves them within the same iteration.

**Requests and routing.** Requests arrive at times $t_j$ with known prompt length $s_j$ and unknown decode length $o_j$ (with prediction $\hat{o}_j$). The router selects a device $g$ and a dispatch period $\tau \geq t_j$ using the same decision variable $x_{j,g,\tau}$.

**Mixed service discipline.** After dispatch, request $j$ joins the FIFO queue of device $g$. When admitted into the active batch, it first undergoes a prefill step (materializing KV states for the prompt) and then executes $o_j$ decode steps. In a PD-mixing engine, a batch at a given period may contain requests in their prefill step and requests in decoding; we treat one such engine iteration as one time period in the model.

**KV-cache memory accounting.** Let $\theta$ denote the service step index of request $j$ since it becomes active on a device, where $\theta = 0$ corresponds to its prefill step and $\theta = 1, \ldots, o_j$ correspond to its decode steps (generating the $\theta$th output token). We approximate that the prefill step allocates the full prompt KV-cache footprint at once. The resulting memory-usage kernel becomes

$$m_j^{\mathrm{mem}}(\theta) = \begin{cases} s_j, & \theta = 0, \\ s_j + \theta, & 1 \leq \theta \leq o_j, \\ 0, & \text{otherwise.} \end{cases}$$

Compared with Section 2, the only change is the additional prefill step $\theta = 0$, whose memory cost is $s_j$. (A finer-grained *chunked prefill* model can be obtained by splitting prefill into multiple steps, but we use the above abstraction for clarity.)

**Feasibility constraints.** The same constraints apply to mixed batches: at any period, each device hosts at most $B$ active requests (counting both prefill and decode), and the total KV-cache usage of active requests must not exceed $M$.

**Token accounting.** In Section 2, all processed tokens are decode tokens. Under PD mixing, devices also process prompt tokens during prefill. When reporting throughput, one may either count only generated (decode) tokens, or count both prompt and generated tokens; both conventions appear in practice. Our formulations and algorithms apply under either convention once the corresponding token count is used consistently.

**Latency metrics under mixing.** Under PD mixing, TTFT and TEL naturally include both prefill and decode waiting/processing time, because the first output token cannot be produced until the request completes its prefill step and begins decoding. The definitions in Section 2 remain valid if $t_j$ is interpreted as the original request arrival time before prefill and $\tau_j$ as the period when the first output token is generated.

**Algorithmic adjustments.** All changes to the online LP algorithm are conceptual substitutions of the decode-only model components by their mixed counterparts. Under the above abstraction, each request consumes $(1 + \hat{o}_j)$ service steps on its assigned device (one prefill step plus $\hat{o}_j$ predicted decode steps), and all per-device capacity constraints are enforced using the modified memory kernel $m_j^{\mathrm{mem}}(\cdot)$.

**Batch-time variability and implications.** A key difference from PD disaggregation is that the wall-clock duration of a mixed batch iteration is input-dependent: it varies with the composition of prefill versus decode work and with prompt lengths. Consequently, it is difficult to map the period-indexed model to real time with a single, accurate conversion factor. Our online LP algorithm can still be applied by treating each iteration as one abstract period and using workload predictions (e.g., $\hat{o}_j$ and $s_j$) for planning; however, because the per-iteration runtime is harder to quantify under PD mixing, the realized performance may be less stable.

# B. Supplementary Materials for Section 3

### B.1. Vectorization: batch-occupancy and memory prices

In this part, we provide the pseudocode of our algorithm.

---

**Algorithm 1** Action-History-Dependent (AHD) Bid Price Control Router

---

1: **Input:** initial resource capacities $b^{(0)} \in \mathbb{R}_+^{|\mathcal{I}|}$ indexed by $i = (g, s, u) \in \mathcal{I}$; horizon $T$; reward/column generator for $(r_{j,g,k}, a_{(j,g,k)})$; weights $(\alpha, \beta, \gamma, \sigma, \zeta_1, \zeta_2)$.

2: **Output:** decisions $\{x_{j,g,k}\}_{k \in [T], j, g}$.

3: Initialize history Hist $\leftarrow \emptyset$ and waiting queue $\mathcal{J}_1^{(wait)} \leftarrow \emptyset$.

4: **for** $k = 1, 2, \ldots, T$ **do**

5:      Update waiting set $\mathcal{J}_k^{(wait)}$ by adding new arrivals at time $k$ and removing any requests started at time $k-1$.

6:      Define action set $\mathcal{A}_k \leftarrow \{(j, g, k) : j \in \mathcal{J}_k^{(wait)}, \ g \in [G]\}$ and initialize $x_{j,g,k} \leftarrow 0$ for all $(j, g, k) \in \mathcal{A}_k$.

7:      **Dual price update (SAA over history):**

$$p_k \in \arg\min_{p \geq 0} \ f_k(p), \tag{9}$$

     where $f_k(p)$ is defined in Equation (6).

8:      **Bid-price margins:** For all $(j, g, k) \in \mathcal{A}_k$, compute $\Delta_j(g) \leftarrow r_{j,g,k} - a_{(j,g,k)}^\top p_k$.

9:      **Update record for next-step learning:** if $\mathcal{A}_k \neq \emptyset$, pick $(j^\star, g^\star) \in \arg\max_{(j,g,k) \in \mathcal{A}_k} \Delta_j(g)$ and set Hist $\leftarrow$ Hist $\cup \{(r_{j^\star, g^\star, k}, a_{(j^\star, g^\star, k)})\}$.

10:      **Greedy admission (bid-price control):** initialize working residuals $b^{(k)} \leftarrow b^{(k-1)}$.

11:      Let $\mathcal{C}_k \leftarrow \{(j, g, k) \in \mathcal{A}_k : \ \Delta_j(g) > 0\}$ sorted by decreasing $\Delta_j(g)$.

12:      **for** each $(j, g, k) \in \mathcal{C}_k$ (in sorted order) **do**

13:          **if** $a_{i,(j,g,k)} \leq b_i^{(k)}$ for all $i \in \mathcal{I}$ **then**

14:              set $x_{j,g,k} \leftarrow 1$ and update $b^{(k)} \leftarrow b^{(k)} - a_{(\cdot,(j,g,k))}$.

15:          **end if**

16:      **end for**

17: **end for**

---

## B.2. Vectorization: batch-occupancy and memory prices

In the fixed-horizon LLM instantiation, constraints are indexed by device $g \in [G]$, time slot $k \in [T]$, and type $u \in \{\mathsf{bat}, \mathsf{mem}\}$. Accordingly, we maintain two nonnegative price tensors

$$p_{g,k}^{\mathsf{bat}} \geq 0, \qquad p_{g,k}^{\mathsf{mem}} \geq 0,$$

and flatten them into a single vector

$$p \equiv \mathrm{vec}(p^{\mathsf{bat}}, p^{\mathsf{mem}}) \in \mathbb{R}_+^{2GT} \equiv \mathbb{R}_+^{|\mathcal{I}|}.$$

**Efficient evaluation of $a^\top p$ for a candidate column.** Consider a candidate action $(j, g, k)$ (start request $j$ on device $g$ at decision step $k$). Its column vector $a_{(j,g,k)}$ has nonzeros only on indices with the same device $g$ and future time slots $k' \geq k$ where the request is active. Let

$$\mathcal{K}_{j,k} := \{\, k' \in [T] : a_j(k' - k) = 1 \,\}$$

denote the active slots of request $j$ if started at $k$. Then the dot product decomposes as

$$a_{(j,g,k)}^\top p \ = \ \sum_{k' \in \mathcal{K}_{j,k}} p_{g,k'}^{\mathsf{bat}} \ + \ \sum_{k' \in \mathcal{K}_{j,k}} p_{g,k'}^{\mathsf{mem}} \cdot \widetilde{m}_j(k' - k), \tag{10}$$

where $\widetilde{m}_j(\cdot)$ is a per-slot memory multiplier derived from $m_j^{\mathsf{mem}}(\cdot)$. In code, we store the active index set $\mathcal{K}_{j,k}$ and the corresponding memory multipliers to avoid materializing the full $2GT$-dimensional column.

**Reward and memory scaling.** To stabilize hinge gradients, the implementation may apply (i) reward normalization and (ii) memory scaling:

$$r^{\mathsf{norm}} := \min\left\{\frac{r}{\mathtt{reward\_normalizer}}, 1\right\}, \qquad \widetilde{m}_j(\cdot) := \mathtt{mem\_scale} \cdot m_j^{\mathsf{mem}}(\cdot),$$

where $\mathtt{mem\_scale}$ is a fixed constant (e.g., proportional to $1/\max_{g,s} M_{g,s}$) to keep the two resource types commensurate.

**Constructing the scarcity vector $d_k$.** At decision step $k$, we form remaining capacities per device and slot:

$$\text{remain}^{\text{bat}} := \left(B - \text{occ}^{\text{bat}}\right)_+, \qquad \text{remain}^{\text{mem}} := \left(M - \text{occ}^{\text{mem}}\right)_+,$$

divide by an estimate of remaining arrivals $\widehat{n}_{\text{remain}}(k)$, and flatten:

$$d_k = \frac{1}{\max\{1, \widehat{n}_{\text{remain}}(k)\}} \text{vec}\Big(\text{remain}^{\text{bat}}, \texttt{mem\_scale} \cdot \text{remain}^{\text{mem}}\Big), \tag{11}$$

(optionally with the same `reward_normalizer` scaling applied as above).

### B.3. Projected mini-batch subgradient routine for the price update

---

**Algorithm 2** RESOLVEDUALPRICES($d_k$, Hist$_k$, $p_{k-1}$): projected mini-batch subgradient for (6)

---

1: **Input:** scarcity vector $d_k \in \mathbb{R}_+^{|\mathcal{I}|}$; history $\text{Hist}_k = \{(r_\ell, a_\ell)\}_{\ell=1}^{N_k}$; warm start $p_{k-1} \in \mathbb{R}_+^{|\mathcal{I}|}$; step size $\eta_k > 0$; steps $K$; batch size $B$.
2: **Output:** updated dual prices $p_k \in \mathbb{R}_+^{|\mathcal{I}|}$.
3: **if** $N_k = 0$ **then**
4:     $p_k \leftarrow 0$.
5:     **return**.
6: **end if**
7: $B \leftarrow \min\{B, N_k\}$.
8: $p \leftarrow p_{k-1}$.
9: **for** $s = 1, 2, \ldots, K$ **do**
10:     Sample $\mathcal{B} \subseteq \{1, \ldots, N_k\}$ uniformly without replacement, $|\mathcal{B}| = B$.
11:     $\widehat{g} \leftarrow d_k$.
12:     **for** each $\ell \in \mathcal{B}$ **do**
13:         Compute margin $\Delta_\ell \leftarrow r_\ell - a_\ell^\top p$ (via (10)).
14:         **if** $\Delta_\ell > 0$ **then**
15:             $\widehat{g} \leftarrow \widehat{g} - \frac{1}{B} a_\ell$.
16:         **end if**
17:     **end for**
18:     $p \leftarrow (p - \eta_k \widehat{g})_+$ {projection onto $\mathbb{R}_+^{|\mathcal{I}|}$}
19: **end for**
20: $p_k \leftarrow p$.
21: **return**.

---

### B.4. How samples enter Hist$_k$ in the online loop

To keep the history compact, we record a small number of informative columns per decision step. A simple choice (used in our simulator) is to record one column corresponding to the largest instantaneous margin under current prices:

$$(j^\star, g^\star) \in \arg\max_{(j,g,k) \in \mathcal{A}_k} \left\{ r_{j,g,k} - a_{(j,g,k)}^\top p_k \right\}, \qquad \text{Hist}_{k+1} \leftarrow \text{Hist}_k \cup \{(r_{j^\star,g^\star,k}, a_{(j^\star,g^\star,k)})\}.$$

This design focuses the dual updates on the most competitive placement options under the current price vector while keeping the per-step learning overhead small.

## C. Supplementary Materials for Section 4

### C.1. Experiments on Real Dataset

**Results under perfect decode-length prediction.** Figures 4, 5, 6 report comparisons between our router and the baseline policies under a stable arrival rate ($\lambda = 0.4$), with objective weights chosen to emphasize (respectively) average latencies, tail latencies, and throughput. Figures 7, 8, 9 report the analogous experiments under a mildly overloaded regime ($\lambda = 0.5$). Across these settings, we make three consistent observations.

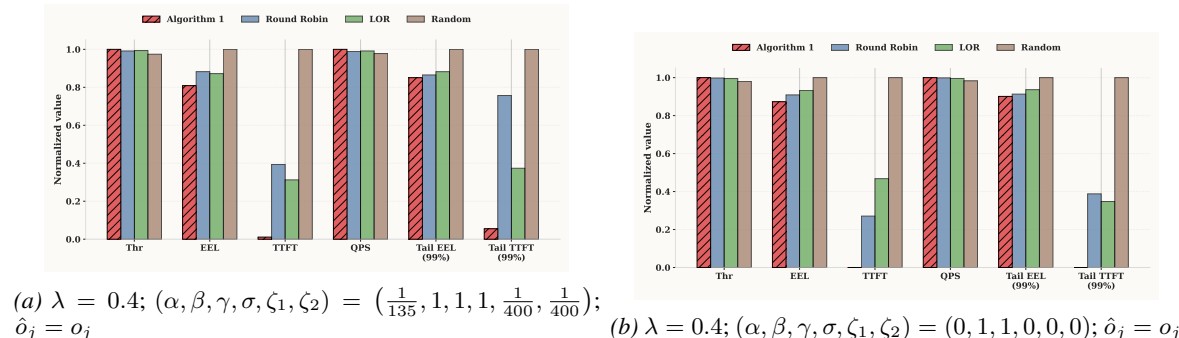

(a) $\lambda = 0.4$; $(\alpha, \beta, \gamma, \sigma, \zeta_1, \zeta_2) = \left(\frac{1}{135}, 1, 1, 1, \frac{1}{400}, \frac{1}{400}\right)$; $\hat{o}_j = o_j$

(b) $\lambda = 0.4$; $(\alpha, \beta, \gamma, \sigma, \zeta_1, \zeta_2) = (0, 1, 1, 0, 0, 0)$; $\hat{o}_j = o_j$

*Figure 4.* Real-data comparison between routing policies under stable arrival process $\lambda = 0.4$, with a focus on average latency.

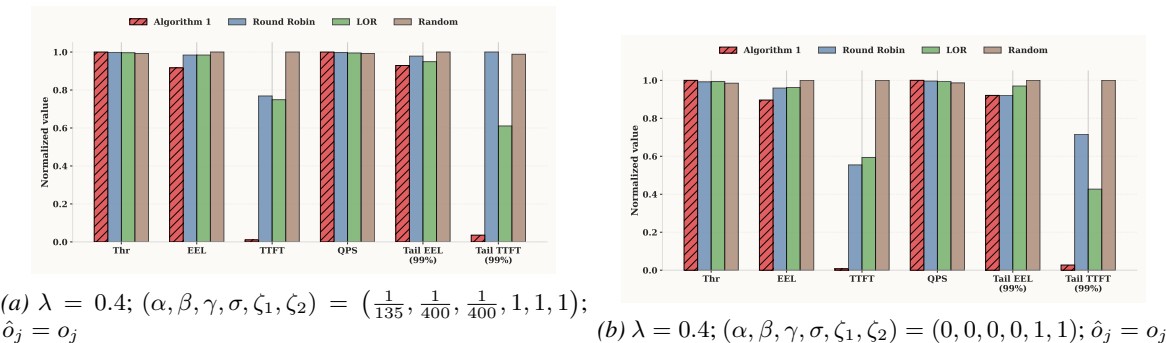

(a) $\lambda = 0.4$; $(\alpha, \beta, \gamma, \sigma, \zeta_1, \zeta_2) = \left(\frac{1}{135}, \frac{1}{400}, \frac{1}{400}, 1, 1, 1\right)$; $\hat{o}_j = o_j$

(b) $\lambda = 0.4$; $(\alpha, \beta, \gamma, \sigma, \zeta_1, \zeta_2) = (0, 0, 0, 0, 1, 1)$; $\hat{o}_j = o_j$

*Figure 5.* Real-data comparison between routing policies under stable arrival process $\lambda = 0.4$, with a focus on tail latency.

1. **Targeted improvements:** when the objective weights emphasize a particular metric family (average latency, tail latency, or throughput), our policy achieves the best performance on that targeted metric compared to all baselines.

2. **Trade-offs are explicit and predictable:** when we set the weights of non-target metrics to zero, the targeted metric typically improves further, but non-target metrics may degrade because they are no longer optimized.

3. **Overload amplifies tail effects:** under $\lambda = 0.5$, tail latencies can deteriorate if tail objectives are assigned insufficient weight. This reinforces the practical importance of explicitly controlling tail-latency terms in overloaded regimes.

**Results under noisy decode-length prediction.** We next evaluate robustness to prediction error by perturbing decode-length estimates via $\hat{o}_j \sim \text{Unif}(0.8\,o_j, 1.2\,o_j)$. Figures 10, 11, 12 present results for the stable regime ($\lambda = 0.4$), and Figures 14, 15, 16 present results for the overloaded regime ($\lambda = 0.5$), again with objective weights emphasizing average latencies, tail latencies, and throughput. The qualitative conclusions mirror the perfect-prediction case: the policy continues to improve the metrics it is asked to optimize, exhibits clear trade-offs when objectives are reweighted, and benefits from explicitly weighting tail terms under overload. Overall, these experiments indicate that the proposed router is robust to moderate decode-length prediction noise.

**Tail weights and the achieved tail quantile.** As discussed in Section 3.1, the tail weights $(\zeta_1, \zeta_2)$ do more than rescale objectives: they directly influence how aggressively the router prioritizes meeting a tail objective. To make this explicit, Figure 13 fixes all parameters except $\zeta_2$, fixes a TTFT threshold $t_2' = 49$, and reports the empirical *satisfaction rate* $q(\zeta_2) := \frac{1}{|\mathcal{I}|} \sum_{j \in \mathcal{I}} \mathbf{1}\{\text{TTFT}_j \leq t_2'\}$, which can be interpreted as the quantile level at which the TTFT threshold $t_2'$ is attained. As $\zeta_2$ increases, $q(\zeta_2)$ increases on average (up to simulation noise), confirming that larger tail weights systematically push the policy toward satisfying a stricter tail requirement for a larger fraction of requests.

### C.2. Experiments on Synthetic Dataset

We complement the real-data experiments with synthetic workloads that control the prefill-to-decode (P/D) ratio, allowing us to stress-test the router under qualitatively different compute/memory regimes. In all cases, we compare against the same

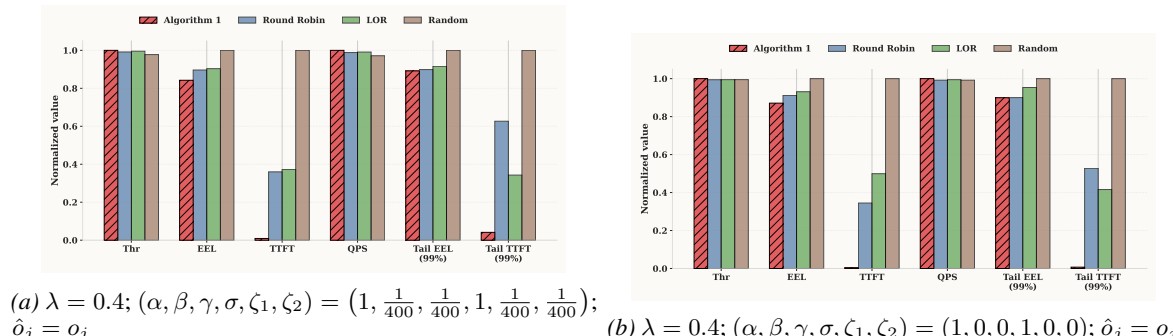

*(a)* $\lambda = 0.4$; $(\alpha, \beta, \gamma, \sigma, \zeta_1, \zeta_2) = \left(1, \frac{1}{400}, \frac{1}{400}, 1, \frac{1}{400}, \frac{1}{400}\right)$; $\hat{o}_j = o_j$

*(b)* $\lambda = 0.4$; $(\alpha, \beta, \gamma, \sigma, \zeta_1, \zeta_2) = (1, 0, 0, 1, 0, 0)$; $\hat{o}_j = o_j$

*Figure 6.* Real-data comparison between routing policies under stable arrival process $\lambda = 0.4$, with a focus on throughput.

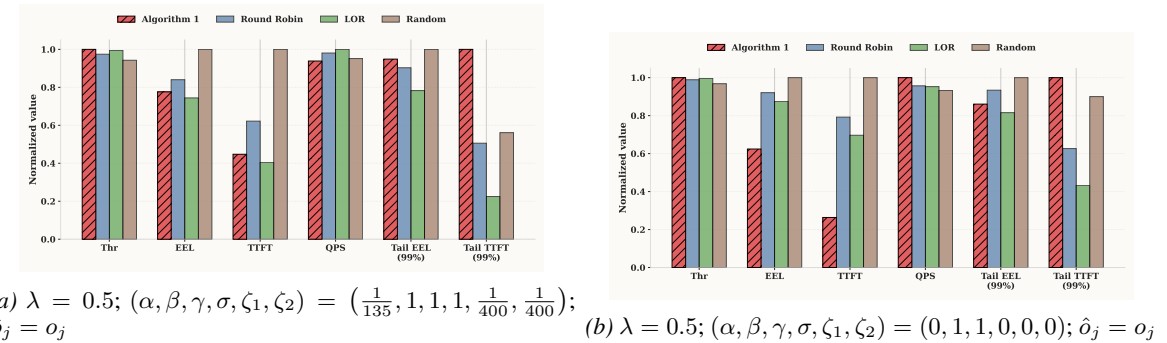

*(a)* $\lambda = 0.5$; $(\alpha, \beta, \gamma, \sigma, \zeta_1, \zeta_2) = \left(\frac{1}{135}, 1, 1, 1, \frac{1}{400}, \frac{1}{400}\right)$; $\hat{o}_j = o_j$

*(b)* $\lambda = 0.5$; $(\alpha, \beta, \gamma, \sigma, \zeta_1, \zeta_2) = (0, 1, 1, 0, 0, 0)$; $\hat{o}_j = o_j$

*Figure 7.* Real-data comparison between routing policies under overloaded arrival process $\lambda = 0.5$, with a focus on average latency.

baseline policies as in the main text and report results under both perfect and noisy decode-length prediction.

**P/D ratio** 1:4 **(decode-heavy).** Figures 17 and 18 report comparisons under a stable arrival rate for decode-heavy requests (P/D = 1:4), with objective weights chosen to emphasize (respectively) average latencies and tail latencies. Figures 19 and 20 report the corresponding experiments under 20% decode-length prediction noise. The qualitative conclusions align with the real-data setting: (i) when the objective weights emphasize a particular metric family, our policy consistently improves that targeted metric relative to the baselines; (ii) reweighting objectives produces predictable trade-offs between average and tail performance; and (iii) the performance advantages persist under moderate prediction noise, indicating robustness of the bid-price control mechanism.

**P/D ratio** 4:1 **(prefill-heavy / decode-light).** Figures 21 and 22 report the analogous comparisons for prefill-heavy requests (P/D = 4:1), again emphasizing (respectively) average and tail latency objectives, with Figures 23 and 24 showing the corresponding noisy-prediction results. In this regime, decode workloads are relatively short, so decode-side batch and KV constraints are less likely to become binding; as a result, small changes in objective weights often lead to smaller changes in observed performance. This highlights an important practical point: the proposed router is most impactful when decode-side congestion is substantial (e.g., long or highly variable decode lengths), where time-coupled batch/KV constraints meaningfully shape the feasible set and where objective-aware admission decisions provide the largest benefit.

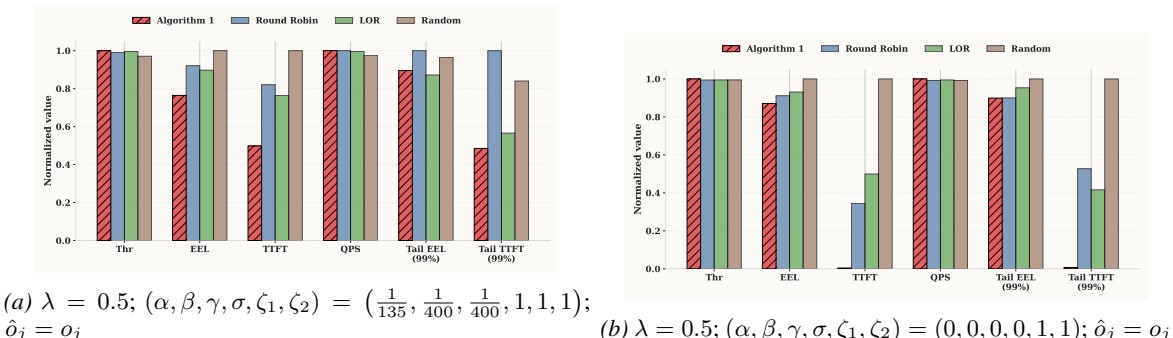

*(a)* $\lambda = 0.5$; $(\alpha, \beta, \gamma, \sigma, \zeta_1, \zeta_2) = \left(\frac{1}{135}, \frac{1}{400}, \frac{1}{400}, 1, 1, 1\right)$; $\hat{o}_j = o_j$

*(b)* $\lambda = 0.5$; $(\alpha, \beta, \gamma, \sigma, \zeta_1, \zeta_2) = (0, 0, 0, 0, 1, 1)$; $\hat{o}_j = o_j$

*Figure 8.* Real-data comparison between routing policies under overloaded arrival process $\lambda = 0.5$, with a focus on tail latency.

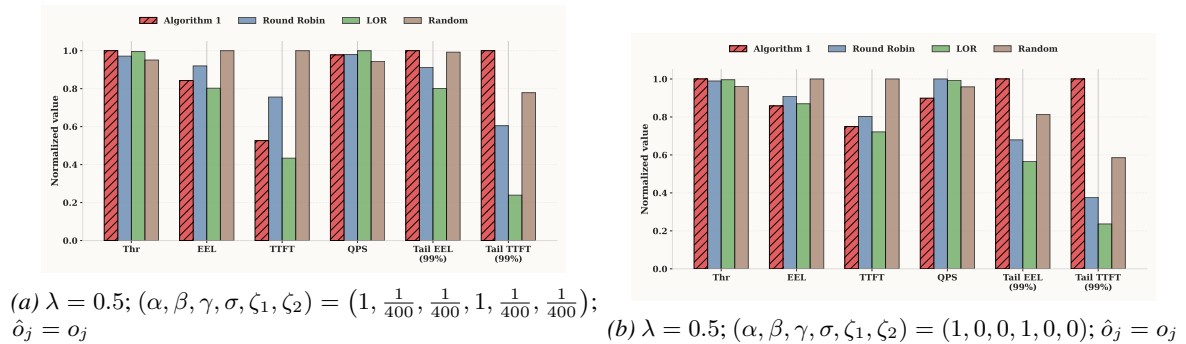

*(a)* $\lambda = 0.5$; $(\alpha, \beta, \gamma, \sigma, \zeta_1, \zeta_2) = \left(1, \frac{1}{400}, \frac{1}{400}, 1, \frac{1}{400}, \frac{1}{400}\right)$; $\hat{o}_j = o_j$

*(b)* $\lambda = 0.5$; $(\alpha, \beta, \gamma, \sigma, \zeta_1, \zeta_2) = (1, 0, 0, 1, 0, 0)$; $\hat{o}_j = o_j$

*Figure 9.* Real-data comparison between routing policies under overloaded arrival process $\lambda = 0.5$, with a focus on throughput.

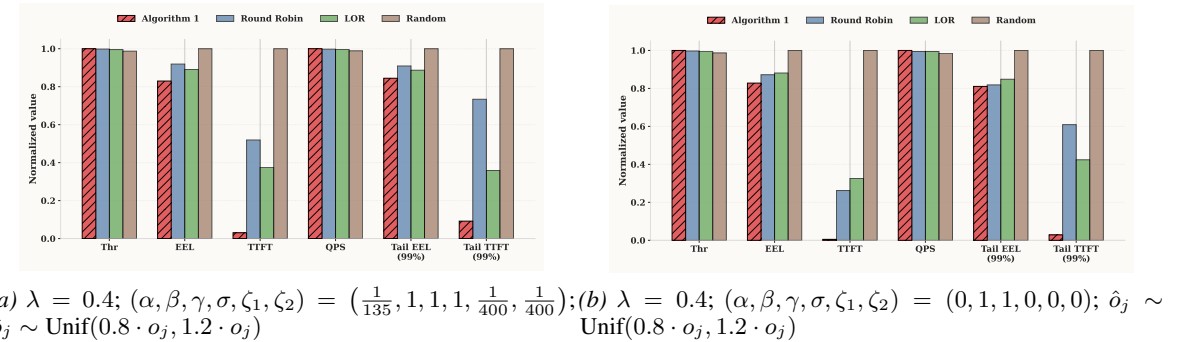

*(a)* $\lambda = 0.4$; $(\alpha, \beta, \gamma, \sigma, \zeta_1, \zeta_2) = \left(\frac{1}{135}, 1, 1, 1, \frac{1}{400}, \frac{1}{400}\right)$; $\hat{o}_j \sim \text{Unif}(0.8 \cdot o_j, 1.2 \cdot o_j)$

*(b)* $\lambda = 0.4$; $(\alpha, \beta, \gamma, \sigma, \zeta_1, \zeta_2) = (0, 1, 1, 0, 0, 0)$; $\hat{o}_j \sim \text{Unif}(0.8 \cdot o_j, 1.2 \cdot o_j)$

*Figure 10.* Real-data comparison between routing policies with noisy decode length prediction under stable arrival process $\lambda = 0.4$, with a focus on average latency.

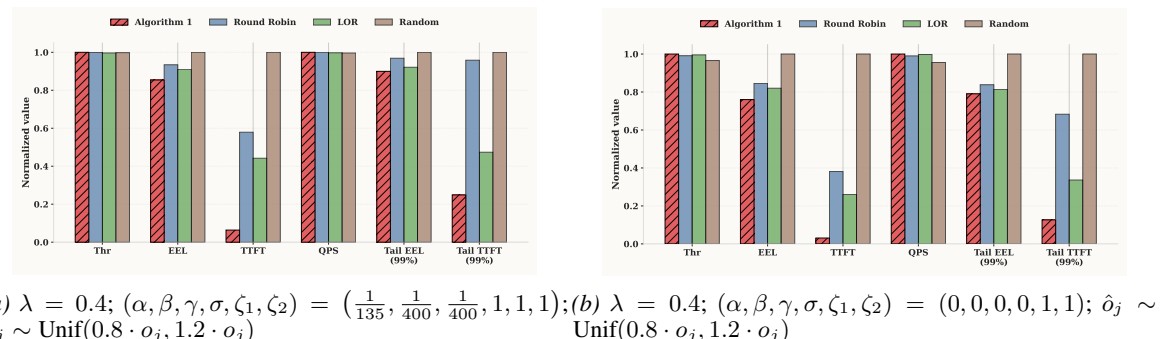

*(a)* $\lambda = 0.4$; $(\alpha, \beta, \gamma, \sigma, \zeta_1, \zeta_2) = \left(\frac{1}{135}, \frac{1}{400}, \frac{1}{400}, 1, 1, 1\right)$; $\hat{o}_j \sim \text{Unif}(0.8 \cdot o_j, 1.2 \cdot o_j)$

*(b)* $\lambda = 0.4$; $(\alpha, \beta, \gamma, \sigma, \zeta_1, \zeta_2) = (0, 0, 0, 0, 1, 1)$; $\hat{o}_j \sim \text{Unif}(0.8 \cdot o_j, 1.2 \cdot o_j)$

*Figure 11.* Real-data comparison between routing policies with noisy decode length prediction under stable arrival process $\lambda = 0.4$, with a focus on tail latency.

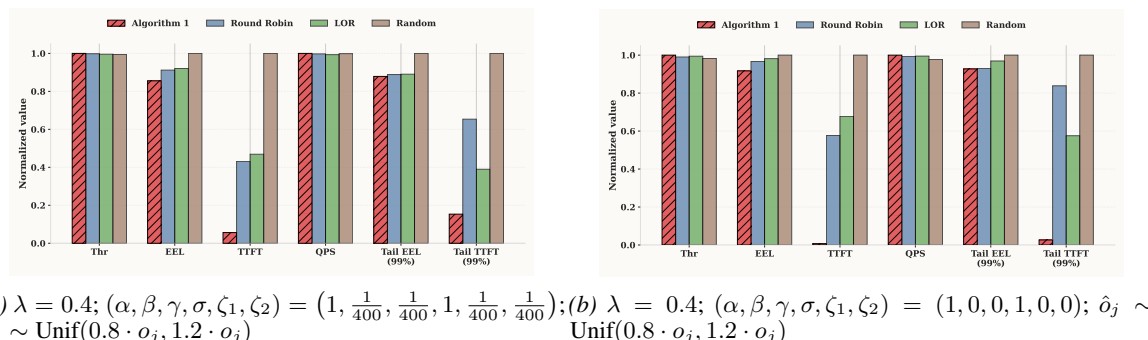

*(a)* $\lambda = 0.4$; $(\alpha, \beta, \gamma, \sigma, \zeta_1, \zeta_2) = \left(1, \frac{1}{400}, \frac{1}{400}, 1, \frac{1}{400}, \frac{1}{400}\right)$; *(b)* $\lambda = 0.4$; $(\alpha, \beta, \gamma, \sigma, \zeta_1, \zeta_2) = (1, 0, 0, 1, 0, 0)$; $\hat{o}_j \sim$
$\hat{o}_j \sim \text{Unif}(0.8 \cdot o_j, 1.2 \cdot o_j)$      $\text{Unif}(0.8 \cdot o_j, 1.2 \cdot o_j)$

*Figure 12.* Real-data comparison between routing policies with noisy decode length prediction under stable arrival process $\lambda = 0.4$, with a focus on throughput.

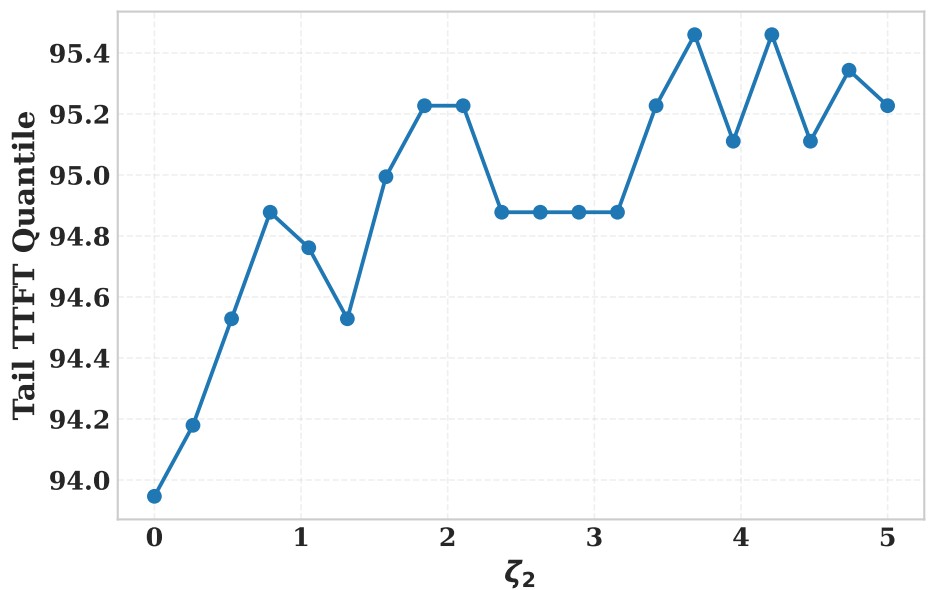

*Figure 13.* TTFT satisfaction rate as a function of the tail weight $\zeta_2$.

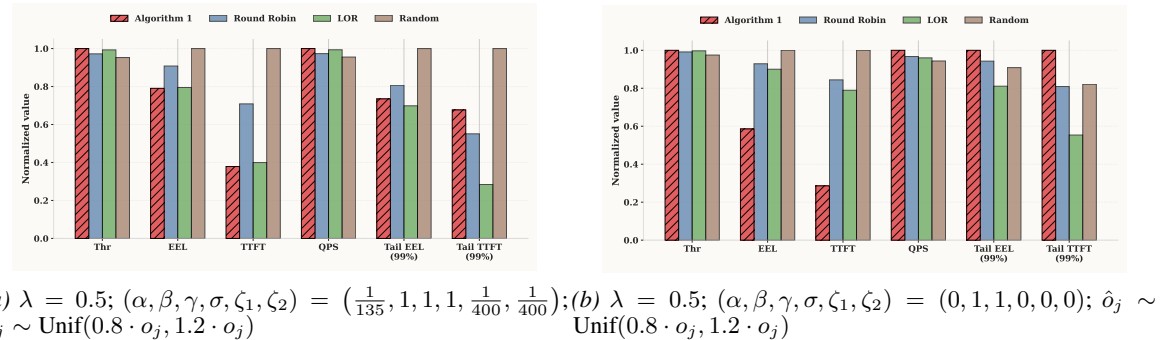

*(a)* $\lambda = 0.5$; $(\alpha, \beta, \gamma, \sigma, \zeta_1, \zeta_2) = \left(\frac{1}{135}, 1, 1, 1, \frac{1}{400}, \frac{1}{400}\right)$; *(b)* $\lambda = 0.5$; $(\alpha, \beta, \gamma, \sigma, \zeta_1, \zeta_2) = (0, 1, 1, 0, 0, 0)$; $\hat{o}_j \sim$
$\hat{o}_j \sim \text{Unif}(0.8 \cdot o_j, 1.2 \cdot o_j)$      $\text{Unif}(0.8 \cdot o_j, 1.2 \cdot o_j)$

*Figure 14.* Real-data comparison between routing policies with noisy decode length prediction under overloaded arrival process $\lambda = 0.5$, with a focus on average latency.

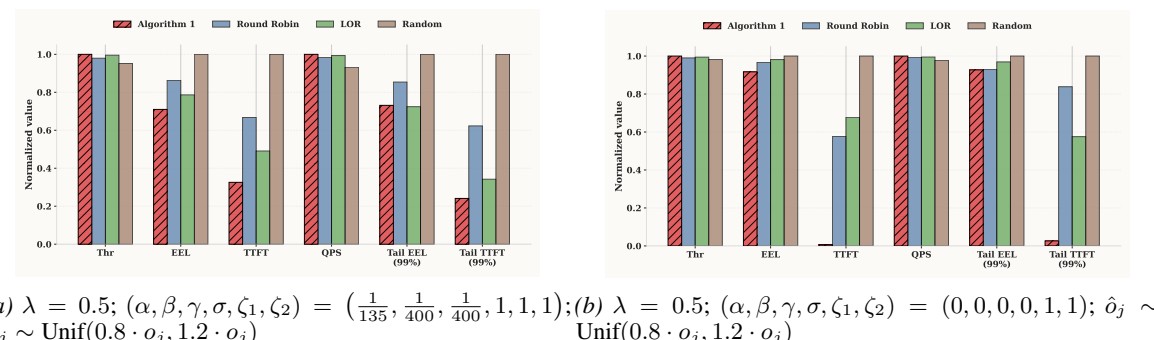

*(a)* $\lambda = 0.5$; $(\alpha, \beta, \gamma, \sigma, \zeta_1, \zeta_2) = \left(\frac{1}{135}, \frac{1}{400}, \frac{1}{400}, 1, 1, 1\right)$; *(b)* $\lambda = 0.5$; $(\alpha, \beta, \gamma, \sigma, \zeta_1, \zeta_2) = (0, 0, 0, 0, 1, 1)$; $\hat{o}_j \sim$ $\hat{o}_j \sim \text{Unif}(0.8 \cdot o_j, 1.2 \cdot o_j)$ $\text{Unif}(0.8 \cdot o_j, 1.2 \cdot o_j)$

*Figure 15.* Real-data comparison between routing policies with noisy decode length prediction under overloaded arrival process $\lambda = 0.5$, with a focus on tail latency.

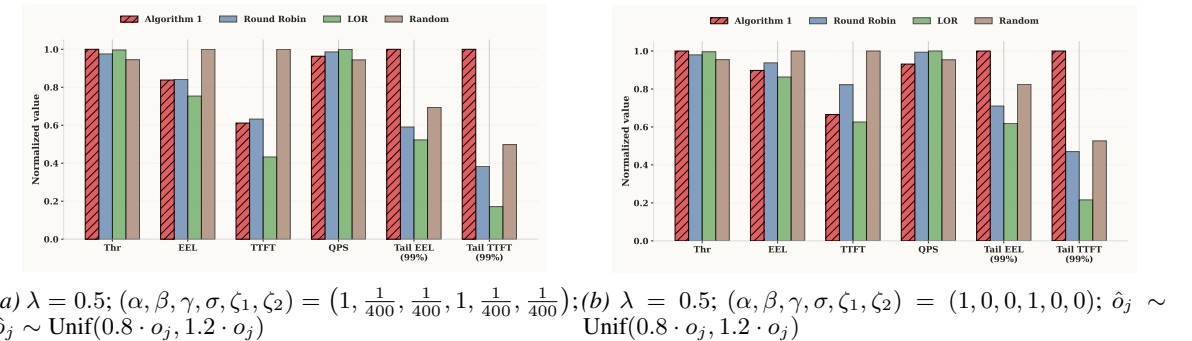

*(a)* $\lambda = 0.5$; $(\alpha, \beta, \gamma, \sigma, \zeta_1, \zeta_2) = \left(1, \frac{1}{400}, \frac{1}{400}, 1, \frac{1}{400}, \frac{1}{400}\right)$; *(b)* $\lambda = 0.5$; $(\alpha, \beta, \gamma, \sigma, \zeta_1, \zeta_2) = (1, 0, 0, 1, 0, 0)$; $\hat{o}_j \sim$ $\hat{o}_j \sim \text{Unif}(0.8 \cdot o_j, 1.2 \cdot o_j)$ $\text{Unif}(0.8 \cdot o_j, 1.2 \cdot o_j)$

*Figure 16.* Real-data comparison between routing policies with noisy decode length prediction under overloaded arrival process $\lambda = 0.5$, with a focus on throughput.

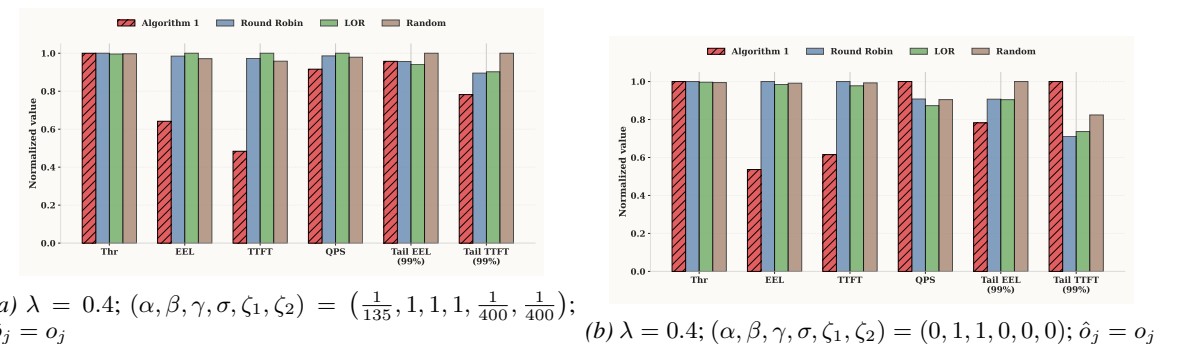

*(a)* $\lambda = 0.4$; $(\alpha, \beta, \gamma, \sigma, \zeta_1, \zeta_2) = \left(\frac{1}{135}, 1, 1, 1, \frac{1}{400}, \frac{1}{400}\right)$; $\hat{o}_j = o_j$

*(b)* $\lambda = 0.4$; $(\alpha, \beta, \gamma, \sigma, \zeta_1, \zeta_2) = (0, 1, 1, 0, 0, 0)$; $\hat{o}_j = o_j$

*Figure 17.* Synthetic data (P/D ratio $1 : 4$) comparison between routing policies under stable arrival process, with a focus on average latency.

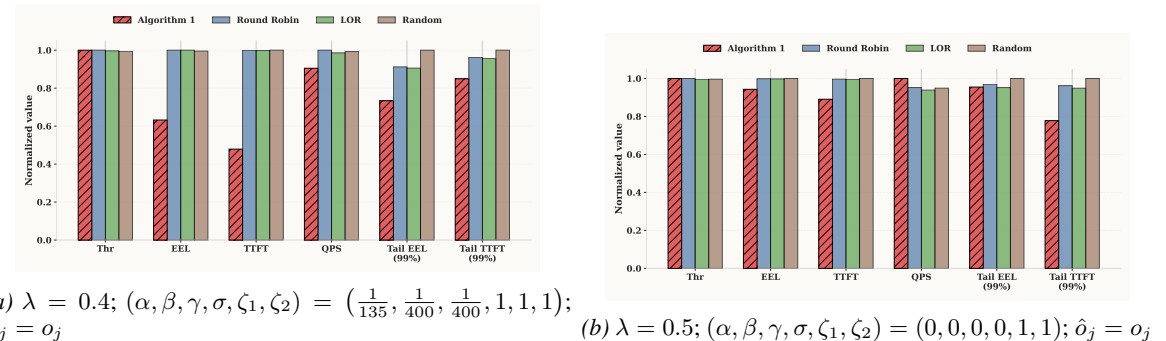

*(a)* $\lambda = 0.4$; $(\alpha, \beta, \gamma, \sigma, \zeta_1, \zeta_2) = \left(\frac{1}{135}, \frac{1}{400}, \frac{1}{400}, 1, 1, 1\right)$; $\hat{o}_j = o_j$

*(b)* $\lambda = 0.5$; $(\alpha, \beta, \gamma, \sigma, \zeta_1, \zeta_2) = (0, 0, 0, 0, 1, 1)$; $\hat{o}_j = o_j$

*Figure 18.* Synthetic data (P/D ratio $1 : 4$) comparison between routing policies under stable arrival process, with a focus on tail latency.

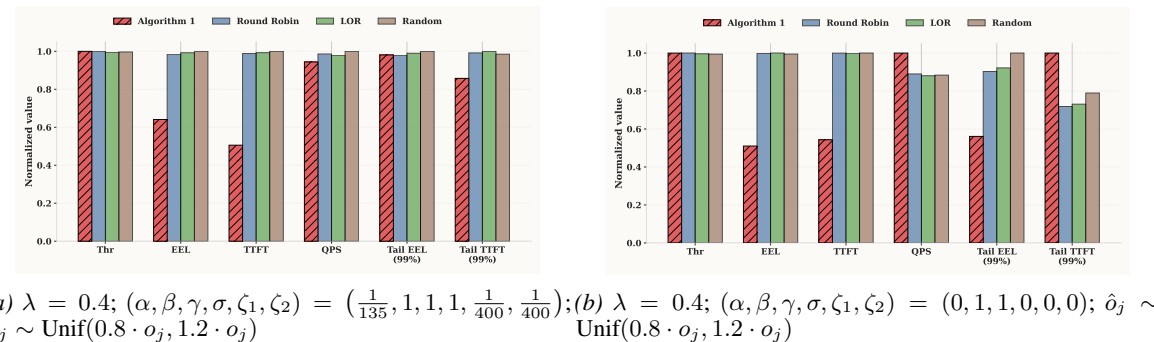

*(a)* $\lambda = 0.4$; $(\alpha, \beta, \gamma, \sigma, \zeta_1, \zeta_2) = \left(\frac{1}{135}, 1, 1, 1, \frac{1}{400}, \frac{1}{400}\right)$;
$\hat{o}_j \sim \text{Unif}(0.8 \cdot o_j, 1.2 \cdot o_j)$

*(b)* $\lambda = 0.4$; $(\alpha, \beta, \gamma, \sigma, \zeta_1, \zeta_2) = (0, 1, 1, 0, 0, 0)$; $\hat{o}_j \sim$ $\text{Unif}(0.8 \cdot o_j, 1.2 \cdot o_j)$

*Figure 19.* Synthetic data (P/D ratio $1:4$) comparison between routing policies with noisy decode length prediction under stable arrival process, with a focus on average latency.

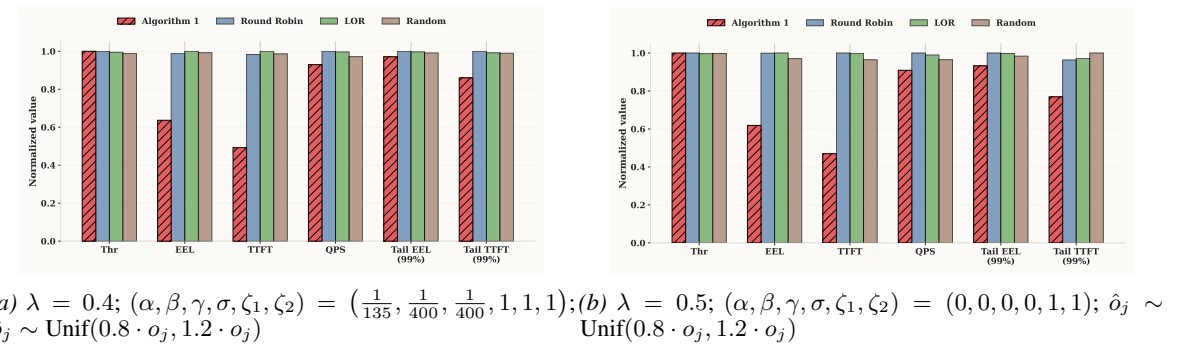

*(a)* $\lambda = 0.4$; $(\alpha, \beta, \gamma, \sigma, \zeta_1, \zeta_2) = \left(\frac{1}{135}, \frac{1}{400}, \frac{1}{400}, 1, 1, 1\right)$;
$\hat{o}_j \sim \text{Unif}(0.8 \cdot o_j, 1.2 \cdot o_j)$

*(b)* $\lambda = 0.5$; $(\alpha, \beta, \gamma, \sigma, \zeta_1, \zeta_2) = (0, 0, 0, 0, 1, 1)$; $\hat{o}_j \sim$ $\text{Unif}(0.8 \cdot o_j, 1.2 \cdot o_j)$

*Figure 20.* Synthetic data (P/D ratio $1:4$) comparison between routing policies with noisy decode length prediction under stable arrival process, with a focus on tail latency.

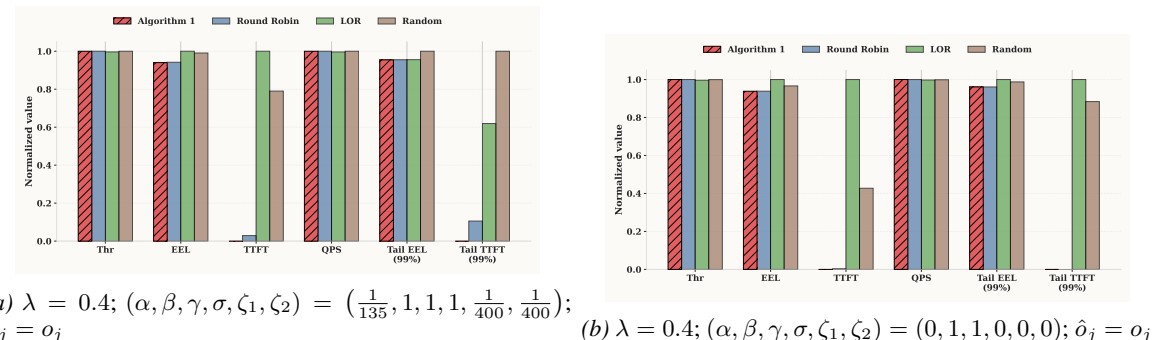

*(a)* $\lambda = 0.4$; $(\alpha, \beta, \gamma, \sigma, \zeta_1, \zeta_2) = \left(\frac{1}{135}, 1, 1, 1, \frac{1}{400}, \frac{1}{400}\right)$;
$\hat{o}_j = o_j$

*(b)* $\lambda = 0.4$; $(\alpha, \beta, \gamma, \sigma, \zeta_1, \zeta_2) = (0, 1, 1, 0, 0, 0)$; $\hat{o}_j = o_j$

*Figure 21.* Synthetic data (P/D ratio $4:1$) comparison between routing policies under stable arrival process, with a focus on average latency.

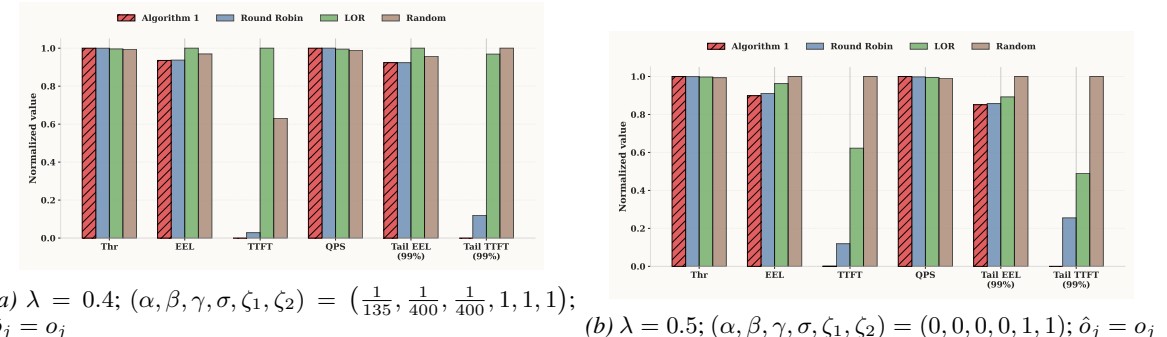

*(a)* $\lambda = 0.4$; $(\alpha, \beta, \gamma, \sigma, \zeta_1, \zeta_2) = \left(\frac{1}{135}, \frac{1}{400}, \frac{1}{400}, 1, 1, 1\right)$;
$\hat{o}_j = o_j$

*(b)* $\lambda = 0.5$; $(\alpha, \beta, \gamma, \sigma, \zeta_1, \zeta_2) = (0, 0, 0, 0, 1, 1)$; $\hat{o}_j = o_j$

*Figure 22.* Synthetic data (P/D ratio $4:1$) comparison between routing policies under stable arrival process, with a focus on tail latency.

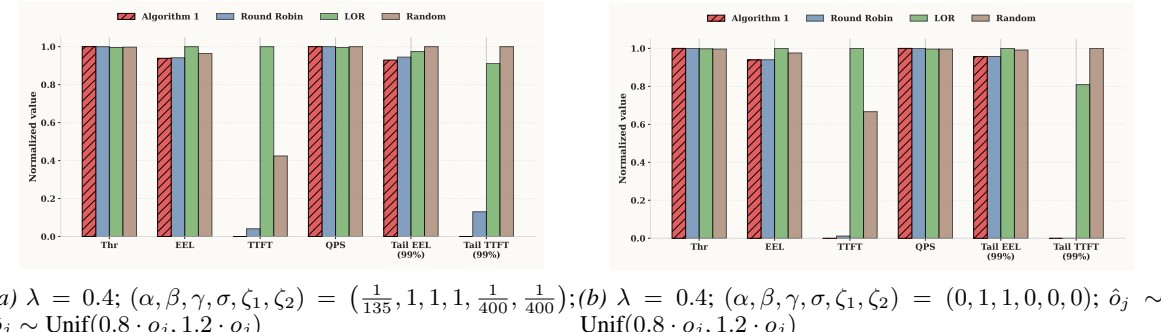

*(a)* $\lambda = 0.4$; $(\alpha, \beta, \gamma, \sigma, \zeta_1, \zeta_2) = \left(\frac{1}{135}, 1, 1, 1, \frac{1}{400}, \frac{1}{400}\right)$; *(b)* $\lambda = 0.4$; $(\alpha, \beta, \gamma, \sigma, \zeta_1, \zeta_2) = (0, 1, 1, 0, 0, 0)$; $\hat{o}_j \sim$ $\hat{o}_j \sim \text{Unif}(0.8 \cdot o_j, 1.2 \cdot o_j)$     $\text{Unif}(0.8 \cdot o_j, 1.2 \cdot o_j)$

*Figure 23.* Synthetic data (P/D ratio $4 : 1$) comparison between routing policies with noisy decode length prediction under stable arrival process, with a focus on average latency.

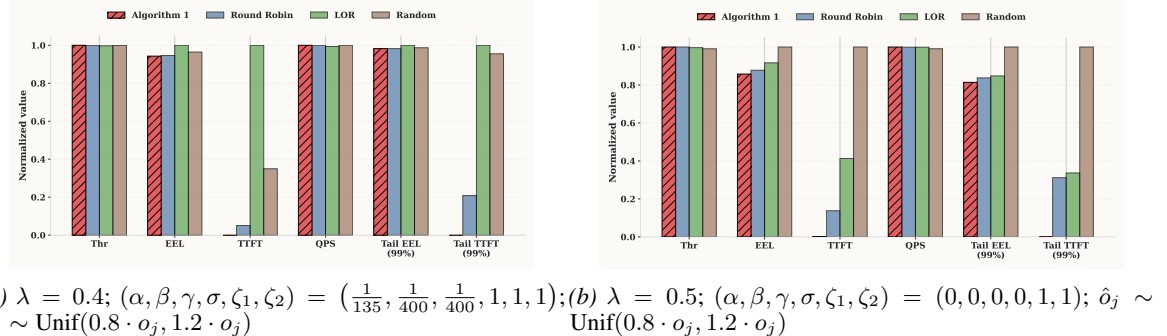

*(a)* $\lambda = 0.4$; $(\alpha, \beta, \gamma, \sigma, \zeta_1, \zeta_2) = \left(\frac{1}{135}, \frac{1}{400}, \frac{1}{400}, 1, 1, 1\right)$; *(b)* $\lambda = 0.5$; $(\alpha, \beta, \gamma, \sigma, \zeta_1, \zeta_2) = (0, 0, 0, 0, 1, 1)$; $\hat{o}_j \sim$ $\hat{o}_j \sim \text{Unif}(0.8 \cdot o_j, 1.2 \cdot o_j)$     $\text{Unif}(0.8 \cdot o_j, 1.2 \cdot o_j)$

*Figure 24.* Synthetic data (P/D ratio $4 : 1$) comparison between routing policies with noisy decode length prediction under stable arrival process, with a focus on tail latency.

