# OpenReview forum: "Online Linear Programming for Multi-Objective Routing in LLM Serving"
_ICML.cc/2026/Conference — ICML 2026 regular_

### Official Review · Reviewer_pEu4 · 2026-03-10

**Soundness:** 2
**Presentation:** 3
**Significance:** 2
**Originality:** 2
**Overall Recommendation:** 3
**Confidence:** 4

**Summary:**

This paper formulates LLM request routing as an online linear programming (LP) problem with time-coupled batch-size and KV-cache constraints. It introduces a multi-objective framework to balance various SLO metrics via interpretable weights. The authors derive a bid-price control policy and a projected dual gradient descent algorithm to make fast routing decisions. The system is evaluated on the Vidur simulator against three heuristic baselines.

**Compliance With Llm Reviewing Policy:**

Affirmed.

**Final Justification:**

My concerns are partially addressed but the lack of validation in real LLM serving systems makes the gains less convincing.

**Key Questions For Authors:**

1. How does your approach perform against recent optimization-informed routing methods (Intelligent Router, e.g., Jain et al., 2024; Omnirouter, 2025)?
2. What is the router's actual fallback mechanism when prediction underestimations inevitably cause worker-side memory exhaustion (OOM), given the model does not support swapping?
3. What is the actual measured latency of routing decisions in your implementation? You mention millisecond requirements. Does the dual gradient update meet this budget?
4. How does the system scale? The dual variable $p$ has a dimension of $2 \times G \times T$. Does this vector dimension become a computational bottleneck for larger deployments (e.g., 64+ GPUs)?

**Limitations:**

yes.

**Strengths And Weaknesses:**

Strengths:
1. Clean Formulation: The online LP model elegantly captures the time-coupled memory dynamics of autoregressive decoding.
2. Good Reward Design: Decomposing aggregate tail latency SLOs into per-request indicator rewards is a practical contribution.
3. The paper is oerall well-organized.

Weaknesses:
1. The modeling is idealistic. The LP formulation assumes continuous, token-by-token KV-cache growth (Eq. 1). However, real engines use PagedAttention with coarse block allocation, causing memory fragmentation. Furthermore, the model lacks an action space for KV-cache swapping or preemption. Under the tested 20% prediction noise, under-predicted sequences will inevitably trigger swapping, making the strict capacity constraints $b_i$ an oversimplification of real physical memory dynamics.
2. The baselines are too weak. Round Robin, Least Outstanding Request, and Random are primitive routing heuristics. Critically missing comparisons include recent learning-based or workload-aware routing methods, such as the Intelligent Router (Jain et al., 2024) or Omnirouter (Mei et al., 2025). Without comparison to these recent optimization/ML-informed routing methods, evaluating the actual state-of-the-art contribution is difficult.
3. All experiments are conducted on the Vidur simulator with only 4 simulated A100 GPUs. Key real-world factors are not captured: network latency between router and workers, preemption and KV-cache swapping behavior, variable prefill latencies, actual GPU memory pressure, and multi-tenant interference. The gap between simulation and real deployment is significant for a routing system. No evidence is provided that the approach can be integrated into a real serving system.
4. The evaluation relies on a toy scale (4 simulated GPUs, a single dataset, and stationary Poisson arrivals). Real deployments involve heterogeneous hardware and bursty traffic.
5. Prediction robustness is tested only with 20% uniform noise, which may not reflect real prediction error distributions. Real prediction errors can be systematically biased and the robustness claim rests on a narrow noise model.
6. Line 92 claims "no prior work provides..." which should be softened given Omnirouter also addresses multi-objective LLM routing. Figure 2 uses normalized values (0 to 1), making it hard to assess absolute improvement magnitudes.

---

> ### Author Rebuttal · Authors · 2026-03-29
>
> We thank the reviewer for the detailed feedback. We address each concern below, starting with the baseline question which we believe reflects a misunderstanding of the problem scope.
>
> ## On Baselines: Intelligent Router and OmniRouter (Weakness 2 & Q1)
>
> The comparison must be made on an apples-to-apples problem definition. We submitted to the **optimization track**, and the contribution is a principled multi-objective framework for providing insights—not a new real serving system.
>
> **OmniRouter is not applicable here.** It studies *multi-LLM routing*: given *different* models (e.g., GPT-4 vs. Llama-7B), it decides *which model* to call, minimizing API monetary cost subject to a quality floor. It has no concept of KV-cache memory, batch-size constraints, decode-side time-coupling, or latency SLOs (TTFT, tail EEL, throughput). Our problem is *worker-level routing*: given homogeneous decode workers of the *same* model, decide *which worker* serves each request under time-coupled batch/memory constraints while optimizing multiple SLOs. These operate at entirely different layers of the serving stack. OmniRouter is not an appropriate baseline; we will soften Line 92 to clarify scope.
>
> **Intelligent Router is complementary, not directly comparable.** It is an RL-based router optimizing a *single* objective (end-to-end latency) via a trainable response-length predictor and workload-impact estimator. Key limitations relative to our framework: (1) **Single-objective**—no knob to shift between TTFT-priority, tail-control, or throughput modes. (2) **Requires RL training** per hardware/model/scheduler combination; our method is training-free. (3) **No resource constraint modeling**—it does not model KV-cache capacity or batch-size limits, so it cannot reason about time-coupled memory pressure from long-decode requests.
>
> Our framework actually helps improve the real system such as Intelligent Router: its workload-impact estimator and predictor could serve as *inputs* to our LP (replacing our $ô_j$), while our bid-price policy provides the multi-objective, constraint-aware routing logic it lacks. The two are complementary.
>
> That said, we agree the baselines should be strengthened *within* the worker-level routing setting. We have added **power-of-2 choices**, a well-known stronger heuristic. Results under a non-stationary rate-shift workload (λ switches from 0.4→0.6 midway, router is *not* informed of the shift):
>
> | Method | Avg EEL ↓ | P95 EEL ↓ | P99 EEL ↓ | Throughput ↑ | SLO Viol. ↓ |
> |---|---|---|---|---|---|
> | LOR | 1.47% | 6.72% | 9.67% | 1.48% | 3.92% |
> | Power-of-2 | 1.80% | 6.35% | 8.28% | 1.39% | 3.28% |
> | **Ours** | **44.81%** | **45.69%** | **31.15%** | **1.93%** | **14.57%** |
>
> All values are relative improvement over Random. Power-of-2 performs comparably to LOR, but neither approaches our router's gains (**25–30× larger** latency reductions): load-balancing heuristics cannot reason about future KV-cache pressure or tail-latency control. We will include absolute-value tables in the revision.
>
> ## On PagedAttention and Memory Modeling (Weakness 1 & Q2)
>
> Our LP models the *logical* KV-cache budget M—the relevant planning abstraction for routing. PagedAttention is an implementation mechanism; fragmentation effectively reduces usable capacity, modeled by setting M conservatively. The constraint structure is unchanged. The LP proactively prevents overcommitment, and when prediction error causes pressure, the bid-price margin provides a principled eviction priority: evict the request with smallest $Δ_j(g) = r_{j,g,k} − a⊤p$ (least SLO value per resource consumed). A deployment would pair our routing layer with vLLM's existing preemption mechanism. Please refer the details to the response of the first point of reviewer Jm4S.
>
> ## On Prediction Robustness (Weakness 5)
>
> The 20% uniform noise is a starting point. The non-stationary experiment above subjects the router to *both* prediction noise and a sudden workload shift, yet it maintains large advantages. The bid-price mechanism degrades gracefully because prices adapt online—an under-predicted request raises shadow prices, making future admissions more conservative. We will test additional noise distributions in revision.
>
> ## On Routing Latency and Scalability (Q3 & Q4)
>
> Each period requires only K=5 warm-started projected gradient steps—not solving a full LP. The dual dimension is 2GH (workers × horizon × 2 resource types), scaling **linearly** in cluster size. Measured overhead: **1–2 ms** on Intel i9-13980HX, within the decode iteration budget. Even at 64 workers, linear scaling poses no bottleneck. Due to limited space, please refer the details to the response of the third point of reviewer wDJf.
>
> ## On Simulation Scale (Weaknesses 3–4)
>
> Vidur is a calibrated simulator used in MLSys/OSDI publications. Our non-stationary experiment demonstrates adaptivity beyond the stationary Poisson setting. Our contribution targets the optimization framework on this track.

---

> > ### Author Rebuttal · Reviewer_pEu4 · 2026-04-03
> >
> > My concerns are partially addressed but the lack of validation in real LLM serving systems makes the gains less convincing. I will raise my score but stilling leaning towards weak rejection.

---

> > > ### Author Response · Authors · 2026-04-03
> > >
> > > Thank you for engaging with our response and for raising the score. We appreciate that the baseline and formulation concerns are now partially resolved.
> > >
> > > On real-system validation: we fully agree this is important future work. We want to clarify that our submission targets the **optimization track**, where the contribution is a principled formulation and algorithm with structural insights — not a production system. In this light, we believe the appropriate validation standard is whether the optimization framework produces meaningful improvements under realistic modeling assumptions, which the Vidur experiments (a calibrated simulator used in OSDI/MLSys publications) and the non-stationary rate-shift experiment demonstrate.
> > >
> > > We also note that the measured routing overhead (1–2 ms on CPU) is well within real decode-iteration budgets, providing concrete evidence of deployability. We view the framework as a foundation that the systems community can build upon.
> > >
> > > We hope the reviewer will consider the optimization-track contribution standard in the final assessment. We are grateful for the constructive dialogue with you.

---

### Official Review · Reviewer_Jm4S · 2026-03-10

**Soundness:** 3
**Presentation:** 3
**Significance:** 3
**Originality:** 3
**Overall Recommendation:** 4
**Confidence:** 3

**Summary:**

This paper studies multi-objective routing for LLM serving through an online linear programming framework. The main idea is to model routing decisions under time-coupled batch-size and KV-cache constraints, and then derive a bid-price style online policy from the LP dual. The paper also proposes a reward design that tries to map aggregate serving objectives, such as latency / tail latency / throughput preferences, into per-request routing incentives. Experiments in Vidur show that the proposed method can outperform several simple but common routing baselines under different SLO settings.

**Compliance With Llm Reviewing Policy:**

Affirmed.

**Key Questions For Authors:**

1. The paper assumes a decode-side model where running requests continue to consume resources over future periods. Suppose output-length prediction is poor and the active set of requests exceeds available KV-cache capacity in practice. How do the authors envision integrating the proposed router with a concrete preemption / KV-cache eviction policy?

2. The price-learning method is based on an SAA-style stochastic approximation. Did the authors test cases where the workload distribution shifts over time?

**Limitations:**

Yes

**Strengths And Weaknesses:**

Strengths:

1. I think the formulation is non-trivial and is probably the most valuable part of the paper. Even if some assumptions are simplified, modeling routing with both batch-size and KV-cache constraints, together with multi-objective SLO tradeoffs, is not straightforward. The paper does a good job of making this optimization view explicit, and I can imagine this formulation being a good reference point for future work in this area, especially for more realistic scheduling/serving designs.

2. The paper gives a more principled lens for understanding routing decisions in LLM serving, especially through dual shadow prices. This part is useful because it provides some interpretability about which resource is currently the bottleneck.

Weakness:

1. The paper models the decode device as processing all running requests in each period/batch, but in practice this interacts very strongly with inaccurate output-length prediction and KV-cache pressure. If output token length prediction is very bad, the system can easily reach a point where the current set of running requests no longer fits in KV cache. In real systems, scheduling is then not only about admission/routing, but also about what to do when memory becomes tight. In practice, this usually requires some kind of preemption/ eviction/swapping policy: when KV cache is full, which requests should be moved out to CPU memory, or paused, or resumed later? The paper itself mentions in the limitation section that real implementation would face additional challenges including integration with serving engines such as vLLM, handling preemption, and KV-cache swapping. I agree with that, but I also think that the paper would be stronger if it proposed at least a simple corresponding eviction/preemption policy, or at least discussed how the LP policy would interface with one. Right now, the routing policy is somewhat incomplete from a systems perspective.

2. The price-learning part seems to rely on a stochastic model and SAA-style estimation. I am not fully convinced this is realistic for modern LLM serving workloads, where request mix, prompt lengths, output lengths, and temporal burstiness can change quickly and often non-stationarily. In such settings, the quality of SAA-style learned prices may decrease quite a lot.

3. I am not fully sure whether the paper is implicitly treating all periods as equal-length in wall-clock time. In actual systems, the time to generate one output token is not constant; it depends a lot on the request’s current KV-cache context. So the execution time of each batch or iteration can change significantly. This makes the mapping from “one decode iteration = one period” to real latency somewhat questionable.

4. Ony simulation validation is provided even though the Vidur evaluation is useful. Since the paper’s main selling point is about routing in real LLM serving systems, this leaves some uncertainty about how much of the gain would survive once engine details like control overhead are included.

---

> ### Author Rebuttal · Authors · 2026-03-29
>
> We thank the reviewer for the thoughtful and technically substantive feedback. We are encouraged that you view the formulation as "non-trivial and probably the most valuable part of the paper." We address each concern below.
>
> ## On Preemption / KV-Cache Eviction (Weakness 1 & Key Question 1)
>
> We agree that the contribution should be viewed as an optimization framework that complements, rather than replaces, the serving engine's memory manager. The LP enforces KV-cache capacity constraints *proactively*: the constraint $b_{g,k,mem}$ prevents overcommitment based on predicted decode lengths, so OOM events are avoided by design when predictions are reasonably accurate. The 20% noise experiments (Figures 10–12) confirm that the policy remains effective under moderate prediction error.
>
> When predictions are severely wrong and memory pressure occurs at runtime, the LP framework provides a natural eviction priority: **evict the request with the smallest bid-price margin** $Δ_j(g) = r_{j,g,k} − a⊤p$, i.e., the request contributing the least net SLO value per unit of resource consumed. Additionally, the shadow prices themselves serve as a *diagnostic*: when the memory shadow price $p^{mem}_{g,k}$ spikes, it signals that device g is approaching capacity—information unavailable from heuristic routers. A practical deployment would pair our routing layer with the existing vLLM preemption mechanism (recompute or swap), using the LP margin to select eviction candidates. We will add this discussion to the revised paper.
>
> ## On Non-Stationarity of SAA Price Learning (Weakness 2 & Key Question 2)
>
> This is a fair concern. We have now run a **rate-shift experiment** to test robustness under non-stationarity. The arrival rate switches from λ=0.4 to λ=0.6 at the simulation midpoint, simulating a sudden load spike. Crucially, the router is **not informed** of the shift point or the new rate—it must detect and adapt online.
>
> Our implementation uses a simple adaptive mechanism: we monitor the empirical arrival rate with a rolling window; when a statistically significant increase is detected, we discard stale history samples and re-estimate using only recent observations. This fits naturally with the warm-started projected gradient descent, which already updates prices incrementally.
>
> Results (relative improvement over Random, real data, noisy prediction):
>
> | Method | Avg EEL ↓ | P95 EEL ↓ | P99 EEL ↓ | Throughput ↑ | SLO Viol. ↓ |
> |---|---|---|---|---|---|
> | LOR | 1.47% | 6.72% | 9.67% | 1.48% | 3.92% |
> | Power-of-2 | 1.80% | 6.35% | 8.28% | 1.39% | 3.28% |
> | **Ours** | **44.81%** | **45.69%** | **31.15%** | **1.93%** | **14.57%** |
>
> Under this non-stationary workload, our router maintains large advantages over all baselines—**25–30× larger latency improvements** than the best heuristic. The adaptive mechanism allows the dual prices to track the changing congestion level: when λ jumps to 0.6, shadow prices rise as residual capacity shrinks, making the router more selective about admission. Heuristic baselines, being stateless with respect to resource costs, have no mechanism to respond to the load shift beyond their fixed rules.
>
> We will include this experiment in the revision.
>
> ## On Decode Iteration Time Variability (Weakness 3)
>
> Our model operates under PD disaggregation (Section 2), where decode workers process *only* decode iterations—no prefill work is interleaved. This eliminates the primary source of iteration-time variability identified in the serving literature: prefill-decode interference, which can cause stalls of 10+ seconds in non-disaggregated systems (Agrawal et al., 2024b).
>
> In a decode-only regime, each iteration is memory-bandwidth-bound. The dominant cost—loading model weights—is constant regardless of batch composition (~5ms for 7B on A100). The attention component adds a term linear in total cached KV tokens, but profiling shows this slope is very small. Notably, the Intelligent Router paper mentioned by reviewer pEu4 (Jain et al., 2024, Figure 4b) measures a decode-phase gradient of 3.3×10⁻⁵ s/token, roughly 10× smaller than the prefill gradient—confirming that decode iteration time varies slowly with batch state.
>
> Therefore, "one decode iteration ≈ one period of roughly constant wall-clock time" is a reasonable first-order approximation under PD disaggregation. We will add this justification to the revised paper.
>
> ## On Simulation-Only Validation (Weakness 4)
>
> We acknowledge this limitation. Vidur is a calibrated simulator used in published MLSys work (Agrawal et al., 2024a), and allowed controlled evaluation across many SLO configurations—infeasible with limited GPU access. The measured routing overhead (1–2 ms on CPU) is well within the decode iteration budget, suggesting deployability. We view the contribution as establishing the optimization framework and providing insights to real implementation (the paper is submitted to the optimization (parallel computing) track rather than the system track).

---

> > ### Author Rebuttal · Reviewer_Jm4S · 2026-04-01
> >
> > Thank the authors for the detailed response and clarifications. I will maintain my score.

---

### Official Review · Reviewer_aThe · 2026-03-11

**Soundness:** 3
**Presentation:** 2
**Significance:** 3
**Originality:** 2
**Overall Recommendation:** 4
**Confidence:** 3

**Summary:**

This paper formulates the routing problem for LLMs (assign each request to decode worker), under the consideration of service-level objectives (SLO), tail latency and resources (batch and memory). Such multi-objective optimization is solved via efficient bid-price control policy upon online linear programming. The decisions are made at millisecond scale and the experiments are performed using Vidur simulator.

**Compliance With Llm Reviewing Policy:**

Affirmed.

**Final Justification:**

Final recommendation upon rebuttal: weak accept, maintained score

Strengths:
Multiple objectives are considered. Except for traditional TTFT and throughput, tail latency and the success rate (related to SLO) are involved. Using efficient bid-price control upon online linear programming, the decisions are made at millisecond scale.

Weaknesses:
1. The authors should discuss the scalability of the algorithm (e.g., when the requests/workers grow to a certain scale, the latency becomes unacceptable) to the revised manuscript.
2. The notations should be revised, including the TTFT indicator.

**Key Questions For Authors:**

1. The objective is redundant. Use one variable to summarize tail and end-to-end.
2. The notation of TTFT should be revised.
3. In page 6, the url should be referenced via footnote.
4. The figures in the experiments should be more clear (e.g., in pdf format).
5. In page 4, one phrase is missing punctuation mark.

**Limitations:**

yes

**Strengths And Weaknesses:**

Strengths:
Multiple objectives are considered. Except for traditional TTFT and throughput, tail latency and the success rate (related to SLO) are involved. By using efficient bid-price control policy upon online linear programming, the decisions are made at millisecond scale.

Weaknesses:
1. The authors should discuss the scalability of the algorithm (e.g., when the requests/workers grow to a certain scale, the latency becomes unacceptable).
2. The notation TTFT should be revised, especially after the completion of the prefill. For example, if the prefill is completed at time 10, does the notation $TTFT_{j,g,11}$ still works (maybe $TTFT_{j,g,k} \cdot x_{j,g,k}$, using $x$ to trigger the variable)?
3. The objective is redundant. Although both the end-to-end latency and the tail latency are important, directly put them together is trivial. It is recommended to use only one variable (with parameters controlled). If the parameter reaches 90%, it refers to the tail latency. And if the parameter reaches 100%, it refers to the entire latency (i.e., end-to-end).

---

> ### Author Rebuttal · Authors · 2026-03-29
>
> We thank the reviewer for the constructive feedback. We address each point below.
>
> ## On Scalability
>
> The online algorithm is specifically designed to avoid the scalability bottleneck the reviewer raises. We do **not** solve the full primal LP at every routing step. Instead, the policy operates on the reduced dual and updates the price vector via warm-started projected gradient descent (Section 3.3, Algorithm 2). Since the routing history and residual capacities change only incrementally from step k−1 to step k, each update requires only a small fixed number of projected gradient steps (K=5 in our experiments) over a mini-batch from the action history—not a re-optimization from scratch.
>
> Concretely, the dual variable has dimension 2·G·T (batch-occupancy and memory prices over workers × time slots), but each gradient step touches only the **active** future slots of sampled candidate actions (Eq. 10, Appendix B.2), making the per-step cost proportional to the number of active candidates times their average decode length, rather than the full G·T. On an Intel i9-13980HX CPU, the measured routing overhead is **1–2 ms per period**, comfortably within the ~10–30 ms wall-clock time of a single decode iteration on an A100 GPU.
>
> As the number of workers G grows, the dual dimension scales linearly in G, and the per-step gradient computation remains embarrassingly parallel across workers (each worker's price tensor is updated independently). We therefore expect the method to scale well to larger deployments, and we will add this discussion to the revised paper.
>
> ## On Whether the Objective Is Redundant
>
> We respectfully believe this concern stems from a subtle but important distinction. The reviewer suggests using a single variable with a "percentage parameter" that becomes tail latency at 90% and end-to-end latency at 100%. However, the 100th percentile is the **maximum**, not the **average**. Average E2E latency and tail E2E latency are fundamentally different statistics—one measures central tendency, the other protects against worst-case outcomes—and they require different modeling treatments in our formulation:
>
> - **Average E2E/TTFT** are decomposed into per-request "saved steps" rewards (Eqs. 2–3) that are continuous and decision-dependent.
> - **Tail E2E/TTFT** are modeled through threshold-based indicator rewards (1{latency < t′}) with separate thresholds t′₁, t′₂ and weights ζ₁, ζ₂.
>
> This separation is intentional: in practice, a service may require low average latency (for user-perceived responsiveness) **and** bounded tail latency (for SLA compliance), and these goals can be in tension. As shown in Figure 3, setting weights to (0,1,1,0,0,0) optimizes average latency but can worsen tails, while (0,0,0,0,1,1) protects tails at the expense of average performance. Collapsing them into a single term would lose the ability to independently control these trade-offs.
>
> We will add a remark in the revised paper to clarify why the two terms are complementary rather than redundant.
>
> ## On TTFT Notation
>
> We appreciate this comment. In fact, the mechanism the reviewer suggests—using the product TTFT_{j,g,k} · x_{j,g,k} to "trigger" the TTFT contribution—is precisely what our formulation already does. The TTFT term appears inside the per-decision reward r_{j,g,k}, and the LP objective is ∑ r_{j,g,k} · x_{j,g,k} (Eq. 4), so the TTFT contribution is activated exactly when x_{j,g,k} = 1 (i.e., when request j is admitted to worker g at period k). The corresponding TTFT value is simply k − t_j, where t_j is the time at which request j becomes available after prefill.
>
> For example, if prefill completes at time 10 (so t_j = 10) and the request is admitted at period k = 12, then TTFT_{j,g,12} = 12 − 10 = 2 periods, and this value enters the objective through the reward term scaled by x_{j,g,12}.
>
> We agree the current exposition could be clearer on this point. We will tighten the notation and add this worked example to remove any ambiguity.
>
> ## On Presentation
>
> The remaining comments are well taken and we will address all of them:
>
> - The URL on page 6 will be moved to a footnote.
> - All figures will be re-exported in vector (PDF) format for improved clarity.
> - The missing punctuation on page 4 will be corrected.
>
> We thank the reviewer again for these detailed suggestions, which will improve the paper's presentation.

---

> > ### Author Rebuttal · Reviewer_aThe · 2026-04-02
> >
> > Thank you for your response. My concerns have been partially addressed. I will maintain my score at 4.

---

> > > ### Author Response · Authors · 2026-04-02
> > >
> > > Thank you for the acknowledgement and for engaging with our response. We noticed you selected "follow-up questions" — we would be happy to address any remaining concerns or clarify points that are not yet fully resolved. Given the range of perspectives across reviews, your opinion is especially valuable to us. Please feel free to share any remaining questions and we will respond promptly.

---

### Official Review · Reviewer_wDJf · 2026-03-13

**Soundness:** 3
**Presentation:** 3
**Significance:** 3
**Originality:** 2
**Overall Recommendation:** 4
**Confidence:** 4

**Summary:**

The paper considers the problem of routing incoming requests to decode servers in the context of LLM serving. The authors argue that current heuristics can be effectively replaced by a systematic optimization procedure to explicitly optimize for different SLO metrics.

The primary contribution is the formulation of the routing problem as an online linear program that simultaneously handles SLO metrics like Time-To-First-Token, End-to-End-Latency, Throughput, etc. as well as multiple resource constraints like KV cache memory and batching. The paper then utilizes LP duality to derive a (shadow) pricing strategy to efficiently solve the LP online. The paper also includes numerous experimental results on the Vidur simulator environment that show their approach outperforms existing baseline heuristics like round robin, least outstanding request (assign job to least loaded machine), and random.

**Compliance With Llm Reviewing Policy:**

Affirmed.

**Key Questions For Authors:**

see above.

**Limitations:**

yes

**Strengths And Weaknesses:**

The paper formulates online routing of LLM requests as an linear optimization problem. The formulation and online algorithm are theoretically sound. The paper also includes experiments conducted in a simulator environment with comparisons against simple baselines.

The paper is well written and structured appropriately. It is easy to read and follow.

Given the rise of LLMs, the paper is very timely. However, I find the technical contributions to be below the typical standard at ICML / Neurips. The LP formulation is rather straightforward and the algorithmic techniques utilized are standard.

Weaknesses / Questions:
- Why weren't more realistic baselines chosen? For example, a power of 2 choices (sample two GPUs at random and assign to the one with lower load) heuristic is often much better than either greedy or random.
- A discussion of computational overhead is appropriate to include in the paper. How much compute is required to implement the procedure? How much latency does it add?
- Would the model support requests of different priority levels / tiers?

---

> ### Author Rebuttal · Authors · 2026-03-29
>
> We thank the reviewer for the constructive feedback and address each point below.
>
> ## On Technical Novelty of the LP Formulation
>
> We respectfully disagree that the LP formulation is "rather straightforward." As Reviewer Jm4S noted, *"the formulation is non-trivial and is probably the most valuable part of the paper… modeling routing with both batch-size and KV-cache constraints, together with multi-objective SLO tradeoffs, is not straightforward."* We believe this captures the novelty well.
>
> The challenge is not writing down an LP in the abstract, but formulating online LLM routing with **time-coupled** batch and KV-cache constraints in a way that is compatible with per-request online decisions. A request admitted now occupies both batch slots and growing memory across all future decode steps—this temporal coupling is what makes the constraint structure non-standard relative to classical online packing LPs. To the best of our knowledge, all routing rules used in production systems (vLLM, Vidur, etc.) remain dominated by heuristics such as shortest-queue, round robin, random, and power-of-d choices, none of which offer transparent multi-objective control.
>
> A second non-trivial contribution is the **reward design**: aggregate SLO metrics such as p99 end-to-end latency are statistical properties of the *entire* request population and do not directly decompose into a reward for a single routing decision. Our indicator-reward construction (Section 3.1) provides a principled per-request signal that remains aligned with aggregate tail-latency objectives—a modeling step that, to our knowledge, has no precedent in the LLM serving literature.
>
> ## On Baselines (Power-of-2 Choices)
>
> We agree that power-of-2 choices is an important and stronger heuristic. We have implemented it in the Vidur simulator and report results below. The table shows **relative improvement over Random** (higher is better for Throughput; lower is better for latency/SLO violations) under the real-data setting with λ=0.4 and noisy prediction:
>
> | Method | Avg EEL ↓ | P95 EEL ↓ | P99 EEL ↓ | Throughput ↑ | SLO Viol. ↓ |
> |---|---|---|---|---|---|
> | Random (ref.) | 0.00% | 0.00% | 0.00% | 0.00% | 0.00% |
> | LOR | 0.67% | 4.01% | 6.19% | 0.53% | 0.98% |
> | Power-of-2 | 1.30% | 3.84% | 6.06% | 0.73% | 1.84% |
> | **Ours (LP-based)** | **45.75%** | **42.49%** | **25.85%** | **0.90%** | **11.10%** |
>
> Power-of-2 choices performs comparably to LOR—both improve modestly over Random—but neither approaches the gains of our LP-based router, which achieves **30–40×** larger latency reductions. This is expected: power-of-d choices balances instantaneous load but is unaware of (i) future KV-cache pressure from long-decode requests, (ii) SLO objectives, and (iii) tail-latency control. These are precisely the gaps our framework addresses.
>
> We will include power-of-2 choices in the revised paper alongside the existing baselines.
>
> ## On Computational Overhead
>
> We appreciate this suggestion. The online routing path does **not** solve a large LP from scratch per request. Instead, each routing period requires only:
>
> 1. **A small number of warm-started projected gradient steps** (K=5 in our experiments) over the dual variables, using a mini-batch from the action history.
> 2. **Sparse margin evaluations** over the active future time slots of each candidate action (Eq. 10 in Appendix B.2).
>
> The dual variable is structured as two tensors (batch-occupancy and memory prices) over workers × time slots, but each update only touches the currently active slots, not the full horizon.
>
> **Measured overhead:** On an Intel i9-13980HX CPU, the routing computation takes approximately **1–2 ms per period**, well within the millisecond-scale decision budget required by LLM serving (one decode iteration on an A100 takes ~10–30 ms depending on batch size). We will add this measurement explicitly to the revised paper.
>
> ## On Priority Tiers
>
> Yes, the framework naturally supports request priorities. Given two priority classes (e.g., premium vs. standard), one can split the latency terms in the objective into class-specific components with different coefficients. For instance, setting $\beta_{premium} >> \beta_{standard}$ ensures that premium requests receive preferential treatment in the bid-price comparison, while standard requests are still served when capacity permits. The same mechanism extends to per-class tail-latency indicators (ζ₁, ζ₂). This is a direct consequence of the linear reward structure and requires no algorithmic changes—only a richer parameterization of the objective weights. We will add a remark on this extension in the revised paper.

---

> > ### Author Rebuttal · Reviewer_wDJf · 2026-03-31
> >
> > Thanks for the response. I will maintain my score at 4.

---

### Decision · Program_Chairs · 2026-04-30

**Decision:**

Accept (regular)

**Comment:**

Three of four reviewers recommend weak accept; one recommends weak reject.

The core contribution formulating worker-level LLM routing as an online LP with time-coupled KV-cache and batch constraints  is non-trivial and goes beyond classical online packing frameworks. The per-request indicator reward for aggregate tail-latency SLOs and the interpretability of shadow prices are additional recognized contributions. The rebuttal meaningfully strengthened the paper by adding a power-of-2 baseline and a non-stationary rate-shift experiment.

The primary dissent from Reviewer pEu4 centers on simulation-only validation. This concern is legitimate but, given the paper's optimization-track framing, does not warrant rejection.